# The TRRAP transcription cofactor represses interferon-stimulated genes in colorectal cancer cells

**Dylane Detilleux, Peggy Raynaud\*, Berengere Pradet-Balade\*, Dominique Helmlinger\***

CRBM, University of Montpellier, CNRS, Montpellier, France

**Abstract** Transcription is essential for cells to respond to signaling cues and involves factors with multiple distinct activities. One such factor, TRRAP, functions as part of two large complexes, SAGA and TIP60, which have crucial roles during transcription activation. Structurally, TRRAP belongs to the phosphoinositide 3 kinase-related kinases (PIKK) family but is the only member classified as a pseudokinase. Recent studies established that a dedicated HSP90 co-chaperone, the triple T (TTT) complex, is essential for PIKK stabilization and activity. Here, using endogenous auxin-inducible degron alleles, we show that the TTT subunit TELO2 promotes TRRAP assembly into SAGA and TIP60 in human colorectal cancer cells (CRCs). Transcriptomic analysis revealed that TELO2 contributes to TRRAP regulatory roles in CRC cells, most notably of MYC target genes. Surprisingly, TELO2 and TRRAP depletion also induced the expression of type I interferon genes. Using a combination of nascent RNA, antibody-targeted chromatin profiling (CUT&RUN), ChIP, and kinetic analyses, we propose a model by which TRRAP directly represses the transcription of *IRF9*, which encodes a master regulator of interferon-stimulated genes. We have therefore uncovered an unexpected transcriptional repressor role for TRRAP, which we propose contributes to its tumorigenic activity.

**\*For correspondence:**
peggy.raynaud@crbm.cnrs.fr (PR);
berengere.pradet-balade@crbm.cnrs.fr (BP-B);
dhelmlinger@crbm.cnrs.fr (DH)

**Competing interest:** The authors declare that no competing interests exist.

## Editor's evaluation

This work will be of wide interest to the transcription community as it is the first evidence that the cofactor, TRRAP, which is known as a transcriptional activator, can also act as a transcriptional repressor. The new experiments added to the revised manuscript further support this conclusion.

## Introduction

Transcriptional regulation is crucial for cells to adapt to external changes, for example, during development or to maintain homeostasis. A critical step in gene expression is the initiation of transcription, which is controlled by both *cis*- and *trans*-regulatory mechanisms. These include the coordinated activities of large, multimeric complexes that modify histones or remodel nucleosomes at promoters. These complexes often function as coactivators, bridging DNA-bound transcription factors to the general transcription machinery. By interacting with many transcription factors, coactivators integrate *cis*-regulatory information from multiple inputs and have important roles in establishing specific gene expression programs. The evolutionary conserved TIP60 and SAGA complexes are two paradigmatic examples of such multifunctional coactivator complexes (*Lu et al., 2009*; *Helmlinger and Tora, 2017*). SAGA carries histone acetyltransferase (HAT) and de-ubiquitination (DUB) activities, which preferentially target histone H3 and H2B, respectively. Similar to the general transcription factor TFIID, with which it shares four core TAF subunits in humans, SAGA delivers TBP to specific promoters, stimulating pre-initiation complex (PIC) assembly. TIP60 also carries HAT activity and, through the EP400

subunit, catalyzes deposition of the H2A.Z variant. TIP60 acetylates histones H4 and H2A, as well as the histone variant H2A.Z.

SAGA and TIP60 share one component, named TRRAP in mammals or Tra1 in yeast (*Grant et al., 1998*; *Vassilev et al., 1998*; *Allard et al., 1999*; *Saleh et al., 1998*). The primary role of Tra1/TRRAP is to recruit SAGA and TIP60 to specific promoters upon binding of an activator. TRRAP was initially discovered as a coactivator for the c-MYC and E2F transcription factors and is essential for their onco-genic activities (*McMahon et al., 1998*). Further work demonstrated that many additional activators require Tra1 or TRRAP to stimulate transcription initiation (*Park et al., 2001*; *Bouchard et al., 2001*; *Lang et al., 2001*; *Deleu et al., 2001*; *Ard et al., 2002*; *Lang and Hearing, 2003*; *Memedula and Belmont, 2003*; *Knutson and Hahn, 2011*; *Lin et al., 2012*). Work in yeast demonstrated that Tra1 physically interacts with the transactivation domain of activators in vivo (*Brown et al., 2001*; *Bhaumik and Green, 2001*; *Bhaumik et al., 2004*; *Fishburn et al., 2005*; *Reeves and Hahn, 2005*). Accord-ingly, genetic studies showed that TRRAP activates the expression of genes involved in a number of important processes. In mammals, these include cell cycle progression (*Herceg et al., 2001*), mitotic checkpoints (*Li et al., 2004*), the maintenance of a pool of stem or progenitor cells (*Loizou et al., 2009*; *Wurdak et al., 2010*; *Tapias et al., 2014*; *Tauc et al., 2017*; *Sawan et al., 2013*), and the regulation of cellular differentiation (*Wang et al., 2018*). However, the direct regulatory roles of Tra1/ TRRAP have been challenging to study because TRRAP is essential for cell proliferation and early embryonic development in mice (*Herceg et al., 2001*). Phenotypic analyses have thus relied on partial depletion and conditional knock-out strategies, which are limited by the slow kinetics and irrevers-ibility of the disruption.

TRRAP is an evolutionary conserved member of an atypical family of kinases, called phosphoinos-itide 3 kinase-related kinases (PIKK), with which it shares a characteristic domain architecture. However, TRRAP lacks all catalytic residues and is therefore the sole pseudokinase of this family (reviewed in *Elías-Villalobos et al., 2019a*). Active PIKKs are implicated in diverse processes. ATM, ATR, and DNA-PKcs control DNA repair and telomere homeostasis, mTOR modulates cell growth, proliferation, and survival in response to metabolic inputs, and SMG-1 mediates nonsense-mediated mRNA decay (*Elías-Villalobos et al., 2019a*; *Lempiäinen and Halazonetis, 2009*). Elegant studies have demon-strated that PIKKs require a dedicated HSP90 co-chaperone, the triple T (TTT) complex, for their stabilization and incorporation into active complexes (*Takai et al., 2007*; *Anderson et al., 2008*; *Takai et al., 2010*; *Hurov et al., 2010*; *Kaizuka et al., 2010*; *Izumi et al., 2010*). TTT was initially discov-ered in the fission yeast *Schizosaccharomyces pombe* and is composed of three conserved, specific subunits, TELO2, TTI1, and TTI2 (*Takai et al., 2010*; *Hayashi et al., 2007*; *Shevchenko et al., 2008*). Biochemical evidence suggests a model by which TTT targets the pleiotropic HSP90 chaperone to PIKKs specifically. Mechanistically, TTT recruits HSP90 through the phosphorylation-dependent inter-action of TELO2 with the R2TP complex, formed by RPAP3, PIH1D1, and the AAA+ ATPases RUVBL1 and RUVBL2 (*Horejsí et al., 2010*; *Hořejší et al., 2014*; *Pal et al., 2014*). Functional studies in various organisms have implicated TTT, particularly TELO2, in PIKK signaling in response to DNA damage or metabolic stress (*Takai et al., 2007*; *Anderson et al., 2008*; *Hurov et al., 2010*; *Kaizuka et al., 2010*; *Ahmed et al., 2001*; *Shikata et al., 2007*; *Izumi et al., 2012*; *Kim et al., 2013*). More recent work showed that TTT itself can respond to signaling cues, such as nutrient levels, to modulate PIKK levels, localization, or substrate binding (*Kim et al., 2013*; *Rao et al., 2014*; *David-Morrison et al., 2016*; *Brown and Gromeier, 2017*). In contrast, the effect of TTT on the incorporation of the TRRAP pseu-dokinase into the SAGA or TIP60 complexes and on their transcription regulatory roles remains poorly characterized, despite evidence that TTT interacts with and stabilizes TRRAP in mammalian cells (*Takai et al., 2007*; *Hurov et al., 2010*; *Kaizuka et al., 2010*; *Izumi et al., 2012*). We recently showed that, in fission yeast, TTT promotes Tra1 stabilization and complex assembly (*Elías-Villalobos et al., 2019b*). Interestingly, however, *S. pombe* apparently lacks orthologs of the R2TP-specific subunits RPAP3 and PIH1D1 (*Inoue et al., 2017*), suggesting that the mechanism of PIKK complex assembly may differ between species. In addition, little is known about how the SAGA and TIP60 complexes are assembled in mammals, which chaperones are required, and whether TRRAP is incorporated into each complex by similar or distinct mechanisms.

Here, we characterized the role of TTT in gene expression and its contribution to TRRAP regula-tory activities. Using CRISPR-Cas9 genome editing, we fused an auxin-inducible degron (AID) to the endogenous TELO2 or TRRAP proteins in human HCT116 colorectal cancer cells (CRCs). Biochemical

and functional analyses indicated that TELO2 controls TRRAP activity in several ways. First, TELO2 promotes the incorporation of TRRAP into both SAGA and TIP60 complexes. Second, TELO2 regulates the expression of a large fraction of TRRAP-dependent genes. Most genes were previously annotated as targets of c-MYC and E2Fs, suggesting that TTT has an important role in sustaining the activities of these oncogenic transcription factors in CRC cells. Unexpectedly, we also found that both TTT and TRRAP inhibit the expression of type I interferon-stimulated genes (ISGs). Taking advantage of the rapid kinetics and reversibility of the degron allele, we provide evidence that TRRAP is a new, direct transcriptional repressor of the *IRF7* and *IRF9* genes, which encode master regulators of ISG induction. To conclude, our work shows that TRRAP, although a pseudokinase, shares a dedicated chaperone machinery with its cognate kinases for its stability and function. Furthermore, we have identified an unexpected repressive role of TRRAP in the transcription regulation of a subset of genes important for innate immunity and resistance to chemotherapy in CRC cells.

## Results

### Rapid and efficient depletion of TELO2 and TRRAP using an auxin-inducible degron

Previous work reported that the TTT components TELO2, TTI1, and TTI2 stabilize TRRAP levels at steady state (*Takai et al., 2007*; *Hurov et al., 2010*; *Kaizuka et al., 2010*; *Izumi et al., 2012*). We therefore sought to determine whether and how TTT contributes to TRRAP-dependent gene regulation. For this aim, we first asked if TTT promotes the incorporation of TRRAP into the SAGA and TIP60 coactivator complexes.

Our initial attempts to knock down TTT components in HCT116 CRC cells using RNA interference (RNAi) produced inconclusive results. We reasoned that, to observe the effects of a chaperone on its clients, we need an inducible depletion strategy that targets protein levels directly. Such systems allow assaying phenotypes rapidly after protein depletion and facilitate the ordering of mechanistic events. In addition, a conditional approach was dictated by the observations that both TELO2 and TRRAP are essential for early embryonic development and cell proliferation (*Herceg et al., 2001*; *Takai et al., 2007*).

For this, we constructed HCT116 cell lines in which an AID sequence is integrated at the endogenous *TELO2* or *TRRAP* loci. The AID system mediates proteasomal-dependent degradation of AID-tagged proteins through auxin-mediated interaction with the F-box protein transport inhibitor response 1 (TIR1) (*Nishimura et al., 2009*; *Holland et al., 2012*). We first generated an HCT116 cell line that stably expresses *Oryza sativa* (OsTIR1). One cell line, which we will refer to as parental control (control), was then used for CRISPR-Cas9-mediated biallelic integration of an AID sequence encoding a full-length IAA17-derived degron (*Figure 1—figure supplement 1A and B*). We also included a YFP fluorescent tag to allow clone selection. We integrated the AID-YFP tag at the 3′-end of *TELO2*. In contrast, we targeted the 5′-end of *TRRAP* because its C-terminal FATC domain is critical for function in yeast (*Hoke et al., 2010*; *Helmlinger et al., 2011*). We also inserted repeated HA epitopes and a 2A peptide (P2A) between the YFP and HA-AID sequences to cleave off the YFP tag during translation in order to lower the risks of affecting TRRAP function with a long fusion sequence. After transfection, sorting, clonal dilution, and amplification, we amplified about 60 clones that showed normal proliferation rates (*Figure 1—figure supplement 1C and D*). We then identified homozygous clones using Western blot with antibodies against endogenous TELO2 and TRRAP by screening for the appearance of a slower-migrating band, indicative of a size shift (*Figure 1—figure supplement 1E and F*). For each tagged protein, three distinct homozygous clones were isolated and further characterized.

We found that auxin addition induces a rapid and robust depletion of either TELO2$^{AID}$ or $^{AID}$TRRAP, whose levels become undetectable after 4 hr of treatment (*Figure 1A and B*). Depletion of either TELO2 or TRRAP progressively decreases the proliferation rate of HCT116 cells (*Figure 1C and D*, *Figure 1—figure supplement 1C and D*), consistent with previous results in mice (*Herceg et al., 2001*; *Takai et al., 2007*). As expected, prolonged TELO2 depletion reduces the steady-state levels of the TRRAP, ATM, ATR, and mTOR proteins and impairs mTORC1 activity, as illustrated with p70-S6K phosphorylation (*Figure 1—figure supplement 1E*). Similarly, prolonged TRRAP depletion reduces the expression of the Cyclin A2 (*CCNA2*) gene (*Figure 1—figure supplement 1F*), which is a known target of TRRAP (*Wurdak et al., 2010*; *Tapias et al., 2014*; *Herceg et al., 2003*). Finally, we verified

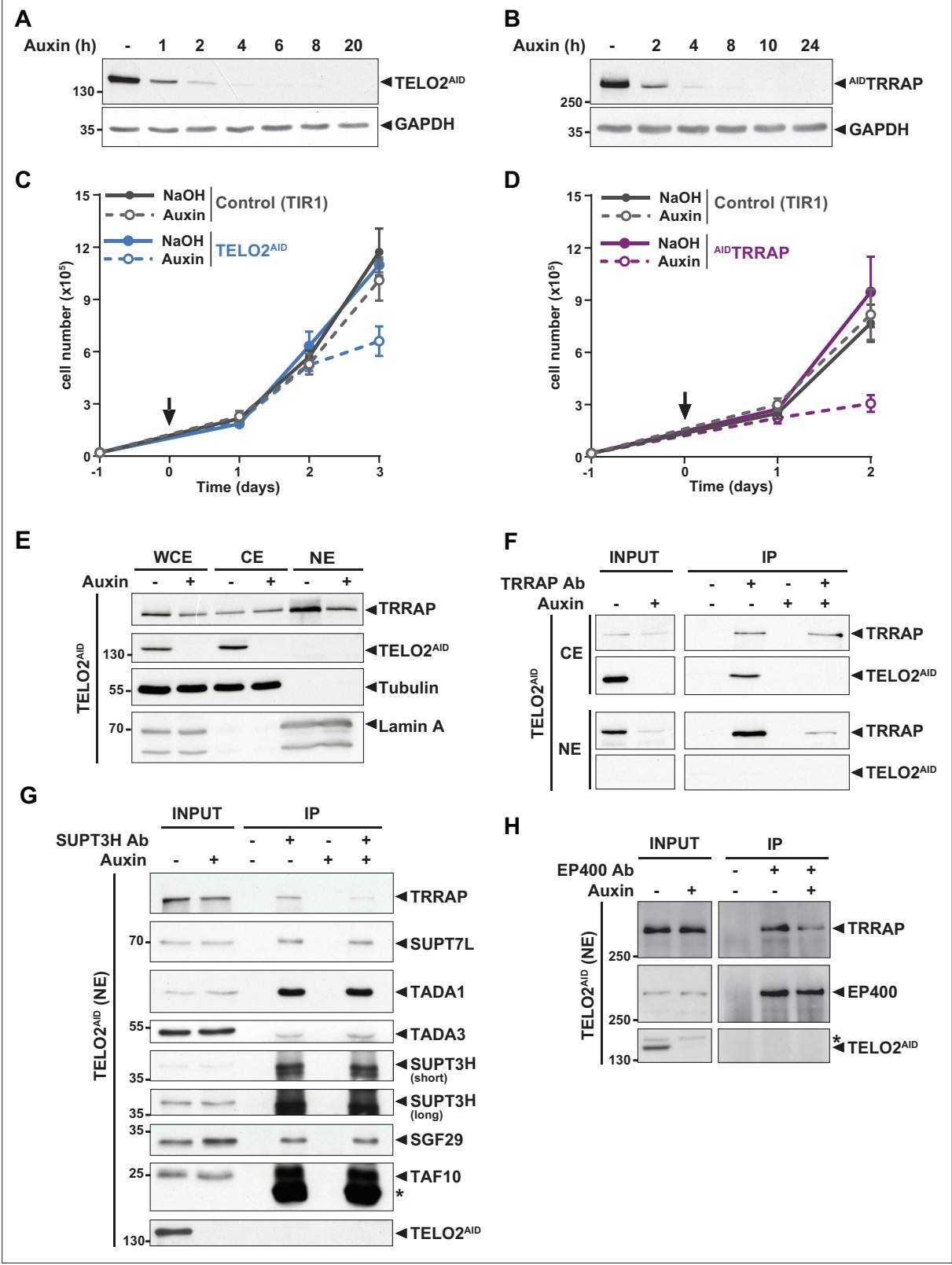

**Figure 1.** TELO2 promotes TRRAP assembly into the SAGA and TIP60 complexes. (**A, B**) Western blot analyses of total extracts from TELO2$^{AID}$ (**A**) or $^{AID}$TRRAP (**B**) cell lines harvested at different time points after auxin addition, as indicated (hours). Blots were probed with an anti-TELO2 antibody (**A**), an anti-HA antibody to detect the HA-AID-TRRAP fusion protein (**B**), and an anti-GAPDH antibody to control for equal loading. (**C, D**) Proliferation rates of parental TIR1 (control, gray lines), TELO2$^{AID}$ (**C**, blue lines), and $^{AID}$TRRAP cells (**D**, purple lines). Cells were seeded 1 day before treatment (arrow)

*Figure 1 continued on next page*

*Figure 1 continued*

with either NaOH (solid lines) or auxin (dashed lines) and counted using trypan blue at the indicated time points. Each value represents average cell counts from four independent experiments, overlaid with the standard deviation (SD). (**E**) Immunoblotting of TRRAP and TELO2$^{AID}$ in whole cell (WCE), cytoplasmic (CE), and nuclear (NE) extracts of TELO2$^{AID}$ cells treated with auxin (+) or NaOH (-) for 48 hr. Tubulin and Lamin A were used as cytoplasmic and nuclear markers, respectively. (**F**) TRRAP immunoprecipitation (Ab +) from CE and NE fractions of TELO2$^{AID}$ cells treated with auxin (+) or NaOH (-) for 48 hr. TRRAP and TELO2$^{AID}$ were revealed by immunoblotting a fraction (2%) of each extract (INPUT) and the entire immunopurified eluate (IP). (**G, H**) SUPT3H and EP400 immunoprecipitation (Ab+) from nuclear extracts of TELO2$^{AID}$ cells treated with auxin (+) or NaOH (-) for 24 hr. SUPT3H (**G**) and EP400 (**H**), TRRAP, and each indicated SAGA subunits were revealed by immunoblotting a fraction (2%) of the extract (INPUT) and the entire IP eluate. Short and long indicate various exposure times. * indicates antibody light chain contamination (**G**) or nonspecific detection (**H**). (**F–H**) Control IPs (Ab-) were performed using beads only. Data are representative of three independent experiments. Source data are available in supplementary material (*Figure 1A–H*, *Source data 1*).

The online version of this article includes the following figure supplement(s) for figure 1:

**Figure supplement 1.** Characterization of TELO2-AID and AID-TRRAP cell lines.

**Figure supplement 2.** TRRAP is not involved in SAGA-dependent endoplasmic reticulum (ER) stress gene induction.

that cell proliferation and PIKK levels are not affected upon treating the parental, TIR1-expressing control cell line with either auxin or its vehicle (*Figure 1C and D*, *Figure 1—figure supplement 1C–F*). In conclusion for this part, we constructed new tools for manipulating TELO2 and TRRAP endogenous levels in a human cell line.

## TELO2 promotes the assembly of TRRAP into the SAGA and TIP60 complexes

We first tested whether TELO2 stabilizes TRRAP at chromatin, where it functions. Cellular fractionation followed by Western blot analysis showed that endogenous TELO2 is detected in cytoplasmic extracts (CE), but not in nuclear extracts (NE) (*Figure 1E*). We observed a similar pattern of the YFP fluorescence signal of TELO2-AID cells using live microscopy (data not shown). In contrast, TRRAP is predominantly nuclear (*Figure 1E*), consistent with its function and published immunofluorescence microscopy staining (*Wurdak et al., 2010*; *Wang et al., 2018*). Nonetheless, TRRAP can also be observed in the cytoplasmic fraction (*Figure 1E*). Immunoprecipitation of TRRAP from each compartment showed that TELO2 interacts specifically with the cytoplasmic fraction of TRRAP, but not the nuclear fraction (*Figure 1F*). We noticed that TELO2 depletion primarily affects nuclear TRRAP levels (*Figure 1E and F*), suggesting that TELO2 binds newly synthesized TRRAP in the cytoplasm to promote its assembly into SAGA and TIP60, which would then be imported into the nucleus.

We therefore examined the effect of TELO2 on TRRAP incorporation into each complex. For this, we treated TELO2$^{AID}$ cells with auxin for only 24 hr, a time point when TRRAP levels remain unchanged (input in *Figure 1G and H*). Immunoprecipitation of the SAGA-specific subunit SUPT3H showed a reduced interaction with TRRAP upon TELO2 depletion (*Figure 1G*). In contrast, TELO2 depletion did not affect the interaction between SUPT3H and the SAGA subunits SUPT7L, TADA1, TADA3, SGF29, and TAF10 (*Figure 1G*). Similarly, we observed a reduced interaction between TRRAP and the TIP60-specific component EP400 upon TELO2 depletion (*Figure 1H*). Overall, we conclude that TELO2 promotes TRRAP stability, nuclear localization, and incorporation into the SAGA and TIP60 complexes. These processes are likely tightly coupled with each other, such that unassembled TRRAP is rapidly degraded and not imported into the nucleus.

## SAGA regulates unfolded protein response genes independently of TRRAP

Interestingly, this analysis also revealed that reducing the levels of TRRAP within SAGA does not affect the interaction between SUPT3H, the bait, and all the other SAGA subunits tested (*Figure 1G*). This observation suggests that TRRAP, despite being about 420 kDa and the largest subunit of SAGA, might be dispensable for SAGA's overall integrity and function, as demonstrated in yeast (*Elías-Villalobos et al., 2019b*; *Helmlinger et al., 2011*). To explore this possibility further, we tested the role of TRRAP in the expression of SAGA-dependent genes during the unfolded protein response (UPR). For this, we monitored UPR gene expression, which requires SAGA for induction in response to endoplasmic reticulum (ER) stress in human cells (*Nagy et al., 2009*; *Lang et al., 2011*). As expected, we found that two such genes, *CHOP* and *HERPUD*, are strongly induced upon treating HCT116

cells with thapsigargin (*Figure 1—figure supplement 2*). Consistent with previous work, their induction decreases about twofold upon RNAi-mediated knockdown of the SAGA core subunit SUPT20H compared with control siRNA transfections. In contrast, despite robust TRRAP depletion (*Figure 1B*), we observed a robust induction of both *CHOP* and *HERPUD* in ^AID^TRRAP cells, irrespective of auxin treatment (*Figure 1—figure supplement 2*). Taken together, these results suggest that TRRAP does not have a major contribution to SAGA integrity and function at these genes. Thus, despite its large size and absence of catalytic activity, TRRAP is presumably not a core subunit of SAGA in human cells, consistent with recent structural studies revealing its peripheral localization within the complex (*Papai et al., 2020*; *Wang et al., 2020*; *Herbst et al., 2021*).

## TELO2 and TRRAP have overlapping regulatory roles in gene expression

We next sought to characterize the role of TTT in gene expression and its contribution to TRRAP regulatory roles. For this, we performed RNA sequencing (RNA-seq) analyses of TELO2- or TRRAP-depleted cells. Based on the growth curves of three distinct clones of each genotype (*Figure 1—figure supplement 1C and D*), we treated TELO2^AID^ and ^AID^TRRAP cells for 48 and 24 hr, respectively. No obvious growth defects were observed at these time points, limiting confounding effects from proliferation arrest. Visualization of aligned reads immediately identified several transcripts whose levels changed in response to TELO2 and TRRAP depletion. For example, expression of *MCIDAS*, a direct target of TRRAP in multiciliated cells (*Wang et al., 2018*), decreases upon TELO2 and TRRAP depletion, while expression of the keratin family gene *KRT20* increases (*Figure 2—figure supplement 1A and B*).

Differential expression analyses revealed both similarities and differences between the transcriptomes of TELO2- and TRRAP-depleted cells. Using a 1% false discovery rate (FDR) cutoff, we identified 2227 transcripts whose levels change upon TRRAP depletion, whereas only 470 transcripts depend on TELO2 (*Supplementary files 1 and 2*). Comparing these two datasets, we found a positive correlation between the gene expression changes caused by TELO2 and TRRAP depletion (Pearson's correlation coefficient $r = 0.66$) (*Figure 2A*). Indeed, a majority of genes whose expression decreases upon TELO2 depletion (152/212) are downregulated in TRRAP-depleted cells (*Figure 2B*). Likewise, almost half of the genes whose expression increases upon TELO2 depletion (118/258) are upregulated in TRRAP-depleted cells (*Figure 2B*). Reverse transcription followed by quantitative PCR (RT-qPCR) analyses confirmed these findings for a few selected genes. We observed that TELO2 and TRRAP promote the expression of the *CCNA2* and *MYB* genes, which encode the cell cycle regulator Cyclin A2 and the Myb transcription factor, respectively (*Figure 2—figure supplement 1E and F*). TELO2 and TRRAP also activate two well-characterized MYC targets genes, *GNL3* and the MiR-17-92a-1 Cluster Host Gene *MIR17HG* (*Figure 2—figure supplement 1E and F*; *Jaenicke et al., 2016*; *Li et al., 2014*). Conversely, both TELO2 and TRRAP inhibit the expression of *KRT20*, a marker of differentiated epithelial cells (*Figure 2—figure supplement 1G and H*). Overall, these observations indicate that TELO2 and TRRAP regulate an overlapping set of genes in HCT116 cells and support our conclusion that TELO2 contributes to TRRAP stability and assembly into SAGA and TIP60 (*Figure 1G and H*).

We then performed an ontology analysis of genes controlled by TELO2 and TRRAP. We used the Gene Set Enrichment Analysis (GSEA) method and the hallmark gene sets from the Molecular Signatures Database (MSigDB) as a reference (*Subramanian et al., 2005*; *Liberzon et al., 2015*). We found that the set of genes downregulated upon TRRAP depletion is mostly enriched for MYC or E2Fs target genes (*Figure 2C*, lower panels). This observation confirms that TRRAP is a coactivator of the c-MYC and E2Fs transcription factors in HCT116 cells, similar to its role in other mammalian cell types. Remarkably, the set of genes activated by TELO2 was also enriched for MYC or E2Fs targets, with comparable enrichment scores (*Figure 2C*, upper panels). Therefore, TELO2 is a novel regulator of the c-MYC and E2Fs transcriptional factors in HCT116 CRC cells, presumably through its role in promoting TRRAP incorporation into the SAGA and TIP60 coactivator complexes.

Despite these similarities, we also observed differences between each transcriptome profile. First, differentially expressed genes are quantitatively more affected upon TRRAP depletion than upon TELO2 depletion (*Figure 2A*, *Figure 2—figure supplement 1C and D*). Second, TRRAP regulates a higher number of genes than TELO2 (*Figure 2B*, *Figure 2—figure supplement 1C and D*). Specifically, most genes that are either downregulated (1125/1277) or upregulated (832/950) upon

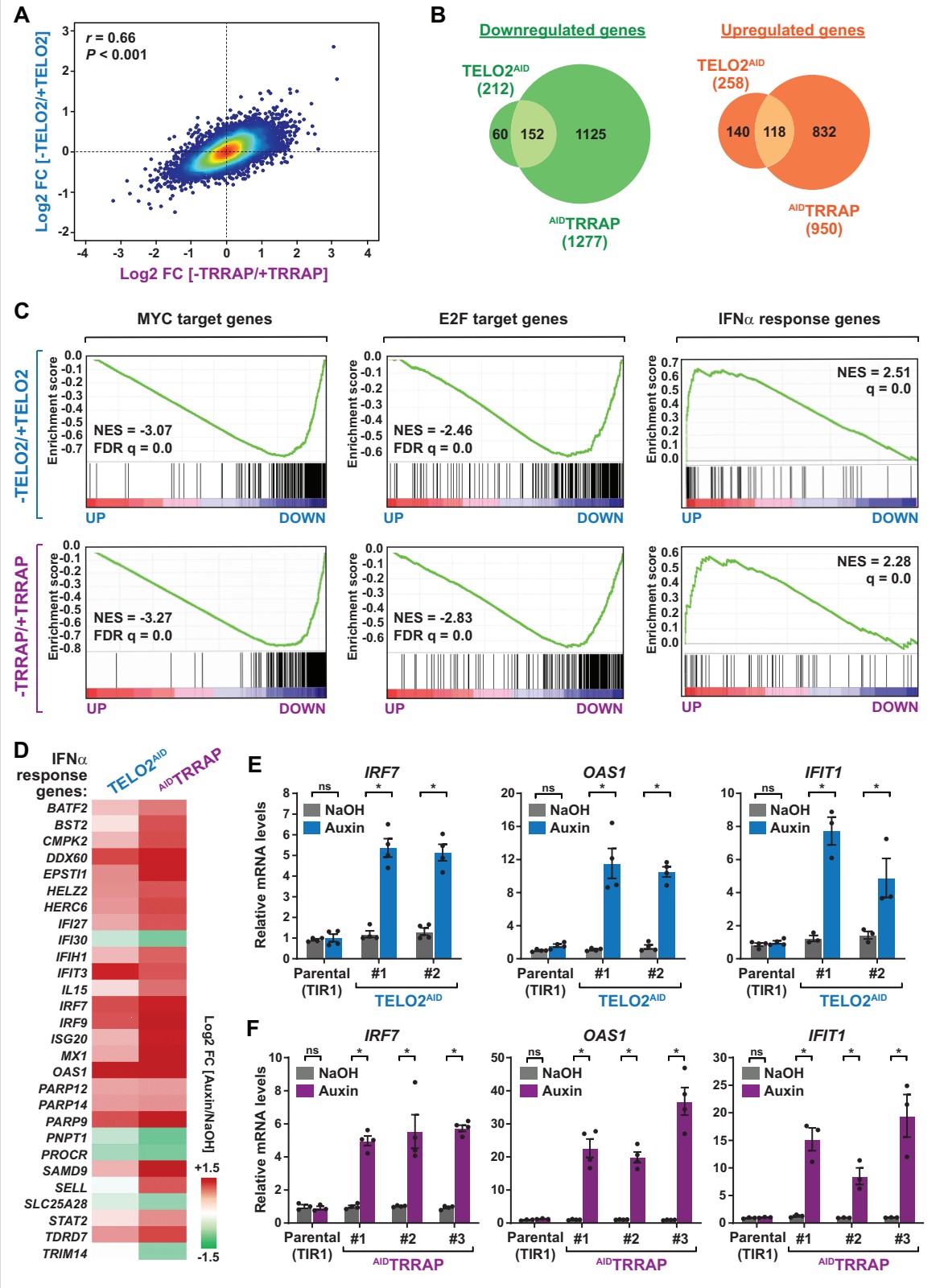

**Figure 2.** TELO2 and TRRAP regulate an overlapping set of genes and repress type I interferon-stimulated genes. (**A**) Density scatter plot comparing gene expression changes between TELO2 and TRRAP-depleted cells. The Pearson correlation coefficient and corresponding p-value are indicated. (**B**) Venn diagrams showing the overlap between genes differentially expressed upon TELO2 and TRRAP depletion (false discovery rate [FDR] ≤ 1%). The number of transcripts whose levels decrease and increase is shown separately, as indicated. (**C**) Gene Set Enrichment Analysis (GSEA) showing

*Figure 2 continued on next page*

*Figure 2 continued*

the most strongly enriched hallmarks in the ranked transcriptome profiles of TELO2- (upper panels) and TRRAP-depleted cells (lower panels). Green lines represent enrichment profiles. NES: normalized enrichment score. Each hit from the hallmark gene set is represented by a vertical black bar, positioned on the ranked transcriptome profile with color-coded fold change values, as indicated. MYC- and E2F-target genes are enriched in the set of genes that are downregulated upon TELO2 and TRRAP depletion, whereas genes from the type I interferon response hallmark are enriched in the set of upregulated genes. (**D**) Heatmap representation of deregulated IFN α-responsive genes in TELO2 and TRRAP-depleted cells (from the 97 genes of hallmark M5911). The Log2 ratio between auxin- and NaOH-treated cells for each transcript is represented using a sequential color scale. All data are from RNA-seq experiments performed in three distinct TELO2$^{AID}$ and $^{AID}$TRRAP clones treated with either auxin or NaOH for 48 and 24 hr, respectively. (**E, F**) Quantitative RT-PCR analysis of three ISGs, *IRF7*, *OAS1*, and *IFIT1*, following the depletion of TELO2 (**E**) and TRRAP (**F**). mRNAs levels were measured in two TELO2$^{AID}$ clones treated with either auxin or NaOH for 48 hr (**E**), and in three $^{AID}$TRRAP clones treated with either auxin or NaOH for 24 hr (**F**). The corresponding parental TIR1-expressing cell lines were also analyzed and treated identically. Each value represents mean mRNA levels from at least three independent experiments, overlaid with individual data points and error bars showing the SD. *PPIB* served as a control for normalization across samples. Values from one NaOH-treated parental control replicate were set to 1, allowing comparisons across culture conditions and replicates. Statistical significance was determined by two-way ANOVA followed by Bonferroni's multiple comparison tests. *p≤0.01.

The online version of this article includes the following figure supplement(s) for figure 2:

**Figure supplement 1.** Gene expression changes upon TELO2 and TRRAP depletion.

**Figure supplement 2.** Auxin treatment induces aryl hydrocarbon receptor (AHR)-responsive genes.

**Figure supplement 3.** TELO2 and TRRAP repress type I interferon-stimulated genes (ISGs).

TRRAP depletion remain unchanged in absence of TELO2 (*Figure 2B*). These differences likely result from the incomplete depletion of TRRAP at this time point of auxin treatment in TELO2$^{AID}$ cells (*Figure 1E and F*, *Figure 1—figure supplement 1E*). Finally, a TORC1 signaling hallmark was specifically enriched in the gene set regulated by TELO2, consistent with its well-characterized roles in TORC1 assembly (*Figure 2—figure supplement 2A*; *Takai et al., 2010*; *Kaizuka et al., 2010*).

## TELO2 and TRRAP repress type I interferon-stimulated genes

Unexpectedly, our ontology analysis also revealed that both TELO2 and TRRAP regulate the expression of genes induced by type I interferons (IFN-I). Using an adjusted p-value threshold of 0.05, we found that nearly one-third of the 97 ISGs (28/97) from the IFNα response hallmark is differentially expressed in the absence of TELO2 and TRRAP (*Figure 2C*, right panels). However, in contrast to the activation of MYC and E2Fs targets, we observed that most ISGs (23/28) are repressed by TELO2 and TRRAP (23/28), whereas they activate the expression of only a few ISGs (5/28) (*Figure 2D*). Accordingly, the ISG response is the most highly enriched category in the set of genes repressed by TELO2 (normalized enrichment score [NES] = 2.51) and TRRAP (NES = 2.28) (*Figure 2C*). To strengthen this observation, we performed RT-qPCR analyses of selected ISGs. We measured the expression of *IRF7*, which encodes a transcription factor acting as a master regulator of IFN-I production and signaling, and of two downstream effectors of this pathway, 2'–5'-oligoadenylate synthetase 1 (*OAS1*) and interferon-induced protein with tetratricopeptide repeats 1 (*IFIT1*) (*Schoggins, 2019*). All three genes exhibit increased mRNA levels upon depletion of either TELO2 or TRRAP (*Figure 2E and F*). Compared to TELO2, we observed that TRRAP depletion causes a stronger induction of ISGs (*Figure 2D–F*).

We next verified that auxin itself does not trigger an IFN-I response. Previous work showed that auxin treatment activates genes regulated by the aryl hydrocarbon receptor (AHR), which is a transcription factor responding to diverse chemicals (*Sathyan et al., 2019*). Analysis of the *cis*-regulatory features enriched in the set of genes repressed by TELO2 or TRRAP identified several genes with AHR-binding motifs in their promoters, confirming this finding in HCT116 cells (*Figure 2—figure supplement 2B*). However, we found that none of the TELO2- or TRRAP-regulated ISGs have AHR-binding motifs within 10 kb of the transcription start sites (TSS), except *IFIT3*. Accordingly, filtering out genes with AHR-responsive motifs did not change the results from the ontology analyses. Finally, we verified that auxin treatment of parental HCT116 cells, expressing TIR1 only, does not affect *IRF7*, *OAS1*, and *IFIT1* expression (*Figure 2E and F*). Altogether, these results show that ISGs are specifically induced in response to TELO2 and TRRAP depletion, indicating that TELO2 and TRRAP repress ISG expression in unstimulated HCT116 CRC cancer cells.

## TRRAP depletion does not activate an innate immune response

TRRAP is best known for its activating role during transcription initiation. Our observation that several ISGs are induced upon TRRAP depletion prompted us to characterize this phenotype further. ISG expression is typically induced in response to pathogens. Infection can trigger various pathogen recognition receptor (PRR) signaling pathways, which converge on TBK1-mediated phosphorylation and activation of the IRF3 and IRF7 transcription factors (reviewed in *Wang et al., 2017*; *Motwani et al., 2019*; *Negishi et al., 2018*). IRF3 and IRF7 stimulate the expression of type I IFNs, particularly the interferon-α and -β cytokines, which are then secreted to activate autocrine or paracrine IFN-I signaling through the JAK/STAT pathway. Phosphorylation of STAT1 and STAT2 promotes the recruitment of IRF9 to form the heterotrimeric transcription factor complex ISGF3, which activates the transcription of ISGs to elicit host defense (*Figure 3A*). Importantly, the *IRF7*, *IRF9*, *STAT1*, and *STAT2* genes are themselves transcriptionally induced by ISGF3 to establish a positive feedback regulatory loop.

To understand how TRRAP regulates ISG expression, we first examined the phosphorylation status of IRF3 and STAT1 in TRRAP-depleted cells as proxies for the activation of the PRR and IFN-I pathways, respectively. Western blot analyses showed no detectable increase in IRF3 Ser396 phosphorylation and STAT1 Ser727 phosphorylation upon TRRAP depletion (*Figure 3B*). IRF3 and STAT1 phosphorylation levels remain unchanged both at early and late time points of TRRAP depletion. We noted that IRF3 and STAT1 protein levels did not change, although RNA-seq showed a modest 1.7-fold increase in STAT1 mRNA levels (*Supplementary file 2*). As a control, we treated ᴬᴵᴰTRRAP cells with a double-stranded (ds) RNA mimic, polyinosinic acid:polycytidylic acid (poly(I:C)), which is a potent IFN-I inducer (*Field et al., 1967*). As expected, poly(I:C) induced IRF3 and STAT1 phosphorylation, validating our experimental conditions and indicating that a functional innate immune response can be triggered in ᴬᴵᴰTRRAP HCT116 cell lines. Therefore, TRRAP depletion induces ISG expression without activating the PRR and IFN-I signaling pathways and presumably acts downstream of the IRF3 and IRF7 transcription regulators.

## Kinetic analysis of ISG regulation by TRRAP

While performing this analysis, we noticed that different innate immune response factors accumulate with apparently distinct kinetics in TRRAP-depleted cells (*Figure 3C and D*). Specifically, the IRF7 and IRF9 transcription factors accumulate earlier than the dsRNA sensor DDX58 (RIG-I). We thus took advantage of the kinetics of the degron system to better characterize the dynamics of ISG expression upon TRRAP depletion in an attempt to order phenotypic events. For this, we used quantitative RT-PCR to follow the expression of representative ISGs over a time course of auxin treatment in ᴬᴵᴰTRRAP cells. This analysis showed that the relative mRNA levels of *IRF9* change rapidly, increasing about twofold already 5 hr after auxin addition (*Figure 3E*), which is concomitant with the loss of a detectable TRRAP signal by Western blot (*Figure 1B*). However, *IRF9* levels do not increase noticeably beyond 10 hr of TRRAP depletion and reach a plateau around fourfold. In contrast, *IRF7* mRNA levels gradually increase over a period of 24 hr of TRRAP depletion and accumulate with a slight delay compared to *IRF9* (*Figure 3E*). Western blot analyses showed the same trend. IRF7 progressively accumulates over the time course of TRRAP depletion, whereas IRF9 begins to level off at an earlier time point, around 15 hr (*Figure 3C and D*). Finally, we found that the expression of ISGF3 targets, including *RIG-I*, *OAS1*, *IFIT1*, and *MX1*, gradually increases upon TRRAP depletion, similar to *IRF7* (*Figure 3E*). Likewise, Western blotting showed that RIG-I upregulation is detectable only 24 hr after auxin addition (*Figure 3D*). Altogether, the progressive accumulation of ISGs indicates that this phenotype appears early after the loss of TRRAP, rather than at late time points. Furthermore, this kinetic analysis indicates that the IRF9 transcription factor is induced earlier than its target genes, including *IRF7*.

## TRRAP represses a specific subset of ISGs

We next sought to determine how TRRAP modulates ISG expression without activating an innate immune response or the IFN-I signaling pathway. Accumulating evidence highlights the existence of non-canonical regulatory mechanisms of ISG transcription. Notably, high levels of IRF9 alone are sufficient to trigger ISG transcription in HCT116 cells (*Kolosenko et al., 2015*). Indeed, IRF9 and STAT2 can form STAT1-independent complexes that drive specific transcriptional programs (*Fink and

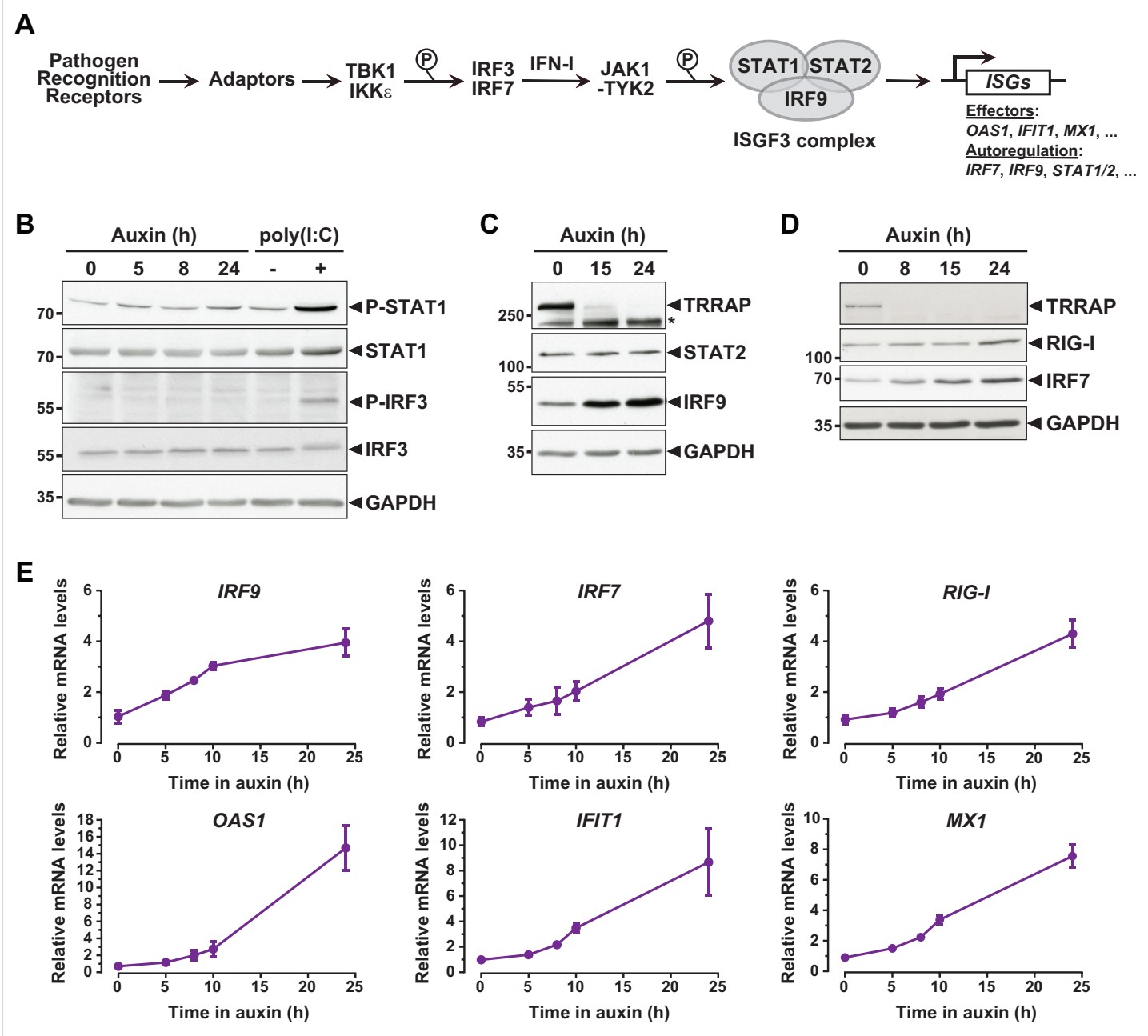

**Figure 3.** Interferon-stimulated gene (ISG) induction in the absence of an innate immune response. (**A**) Schematic representation of the innate immune and IFN-I signaling pathways. (**B**) Western blot analyses of phosphorylated and total STAT1 and IRF3 levels in extracts from ᴬᴵᴰTRRAP cells treated with either NaOH for 24 hr and auxin for various time points, or transfected with polyI:C, as indicated. (**C**) Western blot analyses of TRRAP, STAT2, and IRF9 levels in extracts from ᴬᴵᴰTRRAP cells treated with NaOH for 24 hr or auxin for various time points, as indicated. * indicates nonspecific detection. (**D**) Western blot analyses of TRRAP, RIG-I, and IRF7 levels in extracts from ᴬᴵᴰTRRAP cells treated with NaOH for 24 hr or auxin for various time points, as indicated. (**E**) RT-qPCR analysis of *IRF9*, *IRF7*, *RIG-I*, *OAS1*, *IFIT1*, and *MX1* mRNA levels in ᴬᴵᴰTRRAP cells over a time course of auxin treatment. RNAs were extracted from cells treated with auxin and harvested at various time points, as indicated. Each value represents mean mRNA levels from at least three independent experiments with distinct ᴬᴵᴰTRRAP clones, overlaid with error bars showing the SD for each time point. *PPIB* served as a control for normalization across samples. Values from one untreated replicate were set to 1, allowing comparisons across culture conditions and replicates. Source data are available in supplementary material (**B–D**: *Source data 1*).

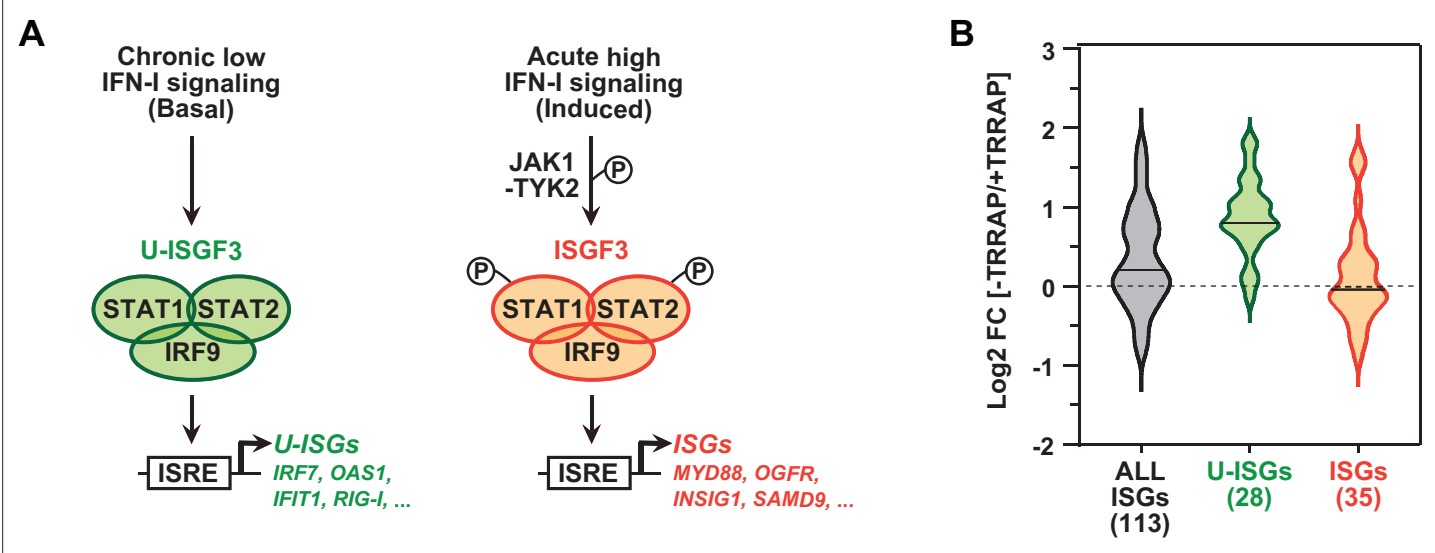

**Figure 4.** TRRAP specifically represses U-ISGF3-regulated genes. (**A**) Schematic representation of the transcription regulation of distinct subsets of interferon-stimulated genes (ISGs), depending on whether ISGF3 unphosphorylated, in basal conditions (U-ISGF3, left), or phosphorylated in response to IFN-I signaling (ISGF3, right). (**B**) Violin plots showing the distribution of gene expression changes induced by TRRAP depletion for all ISGs (gray), for ISGs controlled by U-ISGF3 (green), and for ISGs controlled by ISGF3 (orange). Log2 fold changes (Log2[FC]) were calculated as the Log2 of the ratio of the expression value of each gene between NaOH- (+TRRAP) and auxin-treated (-TRRAP) ^AID^TRRAP cells. The number of genes in each category is indicated.

*Grandvaux, 2014*). Here, our RNA-seq, quantitative RT-PCR, and Western blot analyses indicate that IRF9 levels increase about fourfold (*Supplementary file 2*, *Figure 3C and E*). In contrast, STAT1 and STAT2 levels do not change (*Figure 3B and C*). Thus, TRRAP depletion induces a progressive accumulation of IRF9, but neither activation nor overexpression of its partners STAT1 and STAT2.

Work in both normal and cancer cells demonstrated that IRF9 can assemble with unphosphorylated STAT1 and STAT2 to form the unphosphorylated ISGF3 (U-ISGF3) complex. U-ISGF3 induces the transcription of a subset of 29 ISGs and sustains their prolonged, constitutive expression (*Cheon et al., 2013*; *Sung et al., 2015*; *Platanitis et al., 2019*). In contrast, the phosphorylated form of ISGF3 activates an additional 49 ISGs upon acute IFN-I signaling (*Figure 4A*). We thus compiled a list of 148 ISGs, combining the ISGs from the MSigDB IFNα response hallmark gene set (*Liberzon et al., 2015*) with the U-ISGF3- and ISGF3-regulated ISGs defined in *Cheon et al., 2013*. Of these, 113 ISGs were detected and quantified in our RNA-seq experiments (*Supplementary file 3*). Using an adjusted p-value threshold of 0.05, we found that TRRAP regulates the mRNA levels of about a third of all ISGs (39/113), mostly negatively (30/39) (*Figure 2—figure supplement 3*). Interestingly, we noticed that the effect of TRRAP on all 113 ISGs shows a bimodal distribution (gray violin plot, *Figure 4B*), suggesting that TRRAP regulates a specific subset of ISGs. To compare the effect of TRRAP on U-ISGF3- and ISGF3-regulated genes, we plotted each set separately and observed that TRRAP primarily represses U-ISGF3-dependent genes, while ISGF3-dependent genes remain largely unaffected (compare green and orange violin plots, *Figure 4B*). Specifically, the expression of 20 of the 28 U-ISGF3-dependent genes increases at least 1.5-fold in TRRAP-depleted cells, whereas only 6 of the 35 ISGF3-dependent genes are induced. Finally, we determined whether specific transcription factor binding sites are enriched in the set of 39 ISGs regulated by TRRAP using the i-*cis*Target and Pscan tools (*Zambelli et al., 2009*; *Imrichová et al., 2015*). This analysis identified the interferon-stimulated response element (ISRE) as the most enriched motif in the promoter regions of TRRAP-regulated ISGs (*Figure 2—figure supplement 3*). ISREs are specifically recognized by IRF9 homo- or heterodimers, as well as the ISGF3 complex (*Negishi et al., 2018*), suggesting that IRF9 upregulation is sufficient to induce ISG expression in TRRAP-depleted cells, independently of IFN-I signaling. IRF7-binding sites were the second most enriched motifs in the set of TRRAP-regulated ISGs. In agreement with this observation, IRF7 protein levels progressively accumulate upon TRRAP depletion (*Figure 3D*), consistent with the 2.6-fold increase observed by RNA-seq (*Supplementary file 2*).

In conclusion, TRRAP inhibits the expression of a specific subset of ISGs in HCT116 cells. These genes, hereafter referred to as U-ISGs, are coregulated by U-ISGF3, the unphosphorylated form of the IRF9-STAT1-STAT2 complex, and can be induced independently of activation of the innate immune and IFN-I signaling pathways.

## Genome-wide profiling of TRRAP occupancy

In order to distinguish between direct and indirect regulatory effects, we next asked whether TRRAP binds to the promoter regions of U-ISGs. Although chromatin immunoprecipitation (ChIP) is typically used to identify genomic regions occupied by transcription regulators, this procedure suffers from poor signal-to-noise ratio, particularly for factors that do not bind DNA directly, such as TRRAP and TRRAP-containing complexes. To overcome this problem, we implemented a novel profiling strategy, called Cleavage Under Targets and Release Using Nuclease followed by high-throughput sequencing (CUT&RUN-seq) (*Skene and Henikoff, 2017*; *Meers et al., 2019*). This technique is based on targeted endogenous cleavage of chromatin by micrococcal nuclease (MNase), which is directed to the protein of interest using a specific antibody. Overall, CUT&RUN avoids formaldehyde crosslinking, which can yield misleading results and mask epitopes, and generates very little noise because undigested chromatin and genomic DNA are not extracted.

To benchmark our protocol against the published procedure, we assessed TRRAP occupancy at a well-characterized MYC-bound locus. We reasoned that TRRAP binding reflects that of MYC because they interact both physically and functionally in many cell lines. Accordingly, our RNA-seq analysis shows that TRRAP activates MYC target genes in HCT116 cells (*Figure 2C*). We selected the *MIR17-HG* gene, which encodes an oncomir precursor, is a prominent target of MYC in cancer cells (*Li et al., 2014*), and is activated by TRRAP in HCT116 cells (*Figure 2—figure supplement 1F*). Analysis of published MYC CUT&RUN-seq data confirmed that it binds to a large region of the *MIR17-HG* promoter in human K562 chronic myeloid leukemia cells (*Skene and Henikoff, 2017*). In agreement, using conventional quantitative PCR of DNA fragments released after an anti-MYC CUT&RUN analysis, we found that MYC binds to the promoter of *MIR17-HG* in HCT116 cells (*Figure 5—figure supplement 1A*), validating our protocol. To measure TRRAP binding, we performed CUT&RUN-qPCR using an anti-HA antibody that recognizes the repeated HA epitopes fused to the N-terminal end of ^AIDTRRAP (*Figure 1—figure supplement 1B*). This strategy was remarkably efficient to profile endogenous TRRAP chromatin occupancy (*Figure 5—figure supplement 1B*). Indeed, we observed a strong footprint signal of TRRAP at the promoter region of *MIR17-HG*. Comparison with several negative controls confirmed the robustness and specificity of this observation. First, TRRAP binding at *MIR17HG* is about 60-fold above that observed using control rabbit IgGs. Second, enrichment at the promoter is specific because only background signals are detected downstream, in the gene body. Third, auxin-mediated TRRAP depletion reduces the HA signal to background levels. Interestingly, TRRAP depletion also reduces the binding of MYC to *MIR17-HG*, suggesting that TRRAP contributes to MYC recruitment to promoters (*Figure 5—figure supplement 1A*). We verified that this effect is specific to TRRAP depletion as MYC binding remains unaffected in parental, TIR1-expressing HCT116 cells treated with auxin.

We then measured the genome-wide occupancy of TRRAP using high-throughput sequencing (CUT&RUN-seq). As controls, we performed a similar anti-HA CUT&RUN-seq after auxin-mediated TRRAP depletion and included a rabbit IgG CUT&RUN-seq sample. Visualization of aligned reads immediately identified many TRRAP-dependent footprints on the genome, with a high signal-to noise ratio compared to both control conditions. Statistical analysis of enriched genomic regions using the MACS2 algorithm (*Zhang et al., 2008*) identified 22,219 peaks of TRRAP binding, which decreased down to 849 peaks upon auxin-mediated TRRAP depletion (*Supplementary file 4*). Plotting the distribution of the position of each peak relative to the position of the nearest TSS revealed that TRRAP binds mostly within about 300 base pairs (bp) of the TSS (*Figure 5A*). This range corresponds to the median size of nucleosome-depleted regions (NDRs) measured by NOMe-seq in HCT116 cells (*Lay et al., 2015*), indicating that TRRAP predominantly binds promoter-proximal regions. We conclude that CUT&RUN is an efficient and robust method to profile the genome-wide occupancy of endogenous TRRAP, a component of large chromatin-modifying and -remodeling complexes, in human cells.

Next, we associated each peak to the nearest annotated gene TSS and obtained a list of 11,567 genes that show at least one TRRAP-binding peak (*Supplementary file 4*). To evaluate the relationship

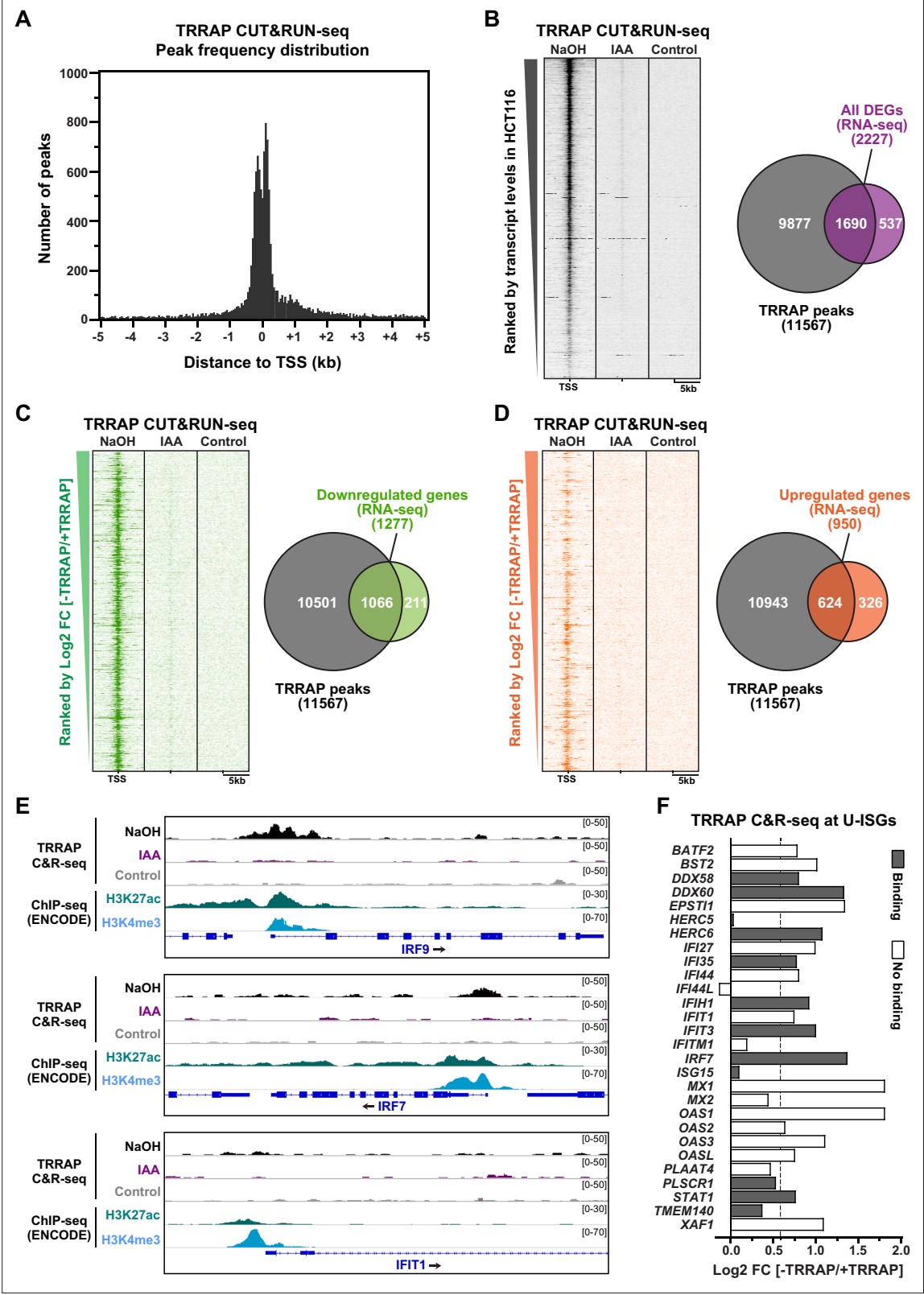

**Figure 5.** Genome-wide profiling of TRRAP occupancy. (**A**) Histogram showing the frequency distribution of TRRAP-bound loci across the genome, relative to annotated transcription start sites (TSS). Shown are the number of peaks computed by MACS2 in bins of 50 bp within ±5 kb of each TSS. (**B–D**) Heatmaps showing anti-HA CUT&RUN-seq profiles of TRRAP, along with control IgG profiles, obtained from ᴬᴵᴰTRRAP HCT116 cells treated with either NaOH or auxin (IAA) for 12 hr, as indicated. For each gene, 5 kb regions upstream and downstream of the TSS are shown and ranked either by

*Figure 5 continued on next page*

*Figure 5 continued*

transcript abundance in control conditions (**B**) or the Log2 fold change (FC) upon auxin-mediated TRRAP depletion (**C, D**). Right panels show Venn diagrams of the overlap between the number of TRRAP-bound promoters and all differentially expressed genes in TRRAP-depleted HCT116 cells (**B**), downregulated genes (**C**), and upregulated genes (**D**) (false discovery rate [FDR] ≤ 1). All transcript-level measurements were obtained from our RNA-seq analyses (*Figure 2*). All anti-HA CUT&RUN-seq experiments were performed in duplicates. (**E**) Scaled snapshots of TRRAP CUT&RUN-seq profiles from ᴬᴵᴰTRRAP HCT116 cells treated as in (**B**) with NaOH (black) and IAA (purple), along with control IgG profiles (gray). Shown are the *IRF9*, *IRF7*, and *IFIT1* loci, from top to bottom. ChIP-seq profiles of the H3K27ac and H3K4me3 marks in HCT116 cells were obtained from the ENCODE database and serve to locate the proximal-promoter regions of each gene. (**F**) Overview of TRRAP binding at promoters of the 28 annotated U-ISGs (*Figure 4*). The bar graph shows the FC of each transcript in TRRAP-depleted HCT116 cells, computed from our RNA-seq analyses (*Figure 2*), color-coded according to either the presence (filled bar) or absence (empty bar) of a TRRAP CUT&RUN peak in their promoters. The dashed line corresponds to a 1.5-fold change threshold.

The online version of this article includes the following figure supplement(s) for figure 5:

**Figure supplement 1.** Implementation of the CUT&RUN procedure.

between TRRAP occupancy and gene expression, we computed a matrix of CUT&RUN-seq read densities for each transcript with a base mean ≥ 5 in our RNA-seq analysis of HCT116 cells (*Figure 2*). Visualization on a heatmap ranked by basal transcript levels demonstrated that TRRAP binds to most genes that are detectably expressed in proliferating HCT116 cells (*Figure 5B*). TRRAP enrichment appears to correlate positively with transcript level, suggesting a global role in RNA polymerase II transcription, as shown for yeast SAGA and NuA4 (*Baptista et al., 2017*; *Bruzzone et al., 2018*). We then computed the overlap between TRRAP binding and effect on gene expression, using a 1% FDR cutoff for differential expression (*Figure 5B*). We found that about three-quarter of TRRAP-regulated genes are bound by TRRAP (1690/2227) and therefore candidates for being direct targets. We then repeated these analyses plotting upregulated and downregulated genes separately (*Figure 5C and D*). Venn diagram and heatmap visualizations showed that a majority of activated genes are bound by TRRAP (83%, 1066/1277) as expected for a transcriptional coactivator. In contrast, a smaller fraction of repressed genes has a TRRAP peak within or near their promoter (66%, 624/950). We noticed that TRRAP appears to bind genes that are modestly induced upon its depletion (*Figure 5D*), suggesting that TRRAP does not have a strong repressive role. Conversely, only 16% of genes (211/1277) are presumably indirectly activated by TRRAP, whereas a higher fraction of genes, about one-third (326/950), are indirectly repressed. To summarize, combining RNA-seq and CUT&RUN-seq analyses with a conditional depletion allele allowed us to identify a total of about 1690 genes that are candidate targets of TRRAP in proliferating HCT116 cells.

## TRRAP binds to the *IRF7* and *IRF9* promoters

We next examined the binding profile of TRRAP at U-ISGs (*Figure 4B*). Visualization of aligned reads identified robust and specific binding of TRRAP to some U-ISGs, including *IRF7* and *IRF9*, but not others, such as *IFIT1* (*Figure 5E*). At both *IRF7* and *IRF9*, comparing the TRRAP CUT&RUN-seq profiles with the ENCODE ChIP-seq profiles of the acetyl H3K27 and trimethyl H3K4 marks confirmed that TRRAP specifically occupies their proximal-promoter regions. Analyzing the overlap between TRRAP binding and regulatory effect revealed that TRRAP binds 12 of the 28 annotated U-ISGs (*Figure 5F* and *Supplementary file 3*). These include eight genes whose expression increases at least 1.5-fold upon TRRAP depletion, suggesting a direct inhibitory effect (*Figure 5F*). Notably, TRRAP binds and represses several genes with pivotal roles in innate immune responses, such as the dsRNA sensor DDX58, also known as RIG-I, as well as the IRF7 and IRF9 transcription factors (*Figure 5E and F*), which mediate the transcriptional induction of downstream effectors of type I IFN signaling. In conclusion, an innovative chromatin profiling technique allowed us to demonstrate that TRRAP directly binds to the promoter regions of some, but not all U-ISGs in HCT116 cells, most importantly *IRF7* and *IRF9*, which encode master regulators of their expression.

## TRRAP represses *IRF7* and *IRF9* transcription

Our results so far suggest that TRRAP depletion induces a specific subset of ISGs in HCT116 cells through, at least in part, increased expression of the IRF7 and IRF9 transcription factors. We thus focused on the regulation of *IRF7* and *IRF9* expression and sought to determine if TRRAP directly represses their transcription.

Our expression analyses were limited to measurements of steady-state mRNA levels, which correspond to the net balance between RNA synthesis and decay. In order to measure the effect of TRRAP on *IRF9* and *IRF7* mRNA synthesis, we performed metabolic labeling of newly synthesized RNAs in TRRAP-depleted cells. For this, ^AID^TRRAP cells were treated with auxin for ten hours, followed by a short incubation with 4-thiouridine (4sU). 4sU-labeled RNAs were then purified and quantified by conventional quantitative RT-PCR. To verify the efficiency of labeling and purification, we amplified intronic regions of each gene. We found that the levels of newly synthesized *IRF7* and *IRF9* pre-mRNAs increase about 1.8- and 1.6-fold in the absence of TRRAP, respectively (*Figure 6A*). Upon TRRAP depletion, we observed an eightfold increase in *OAS1* transcription as expected for a target of the IRF7 and IRF9 activators. We conclude that TRRAP inhibits *IRF7* and *IRF9* mRNA synthesis, suggesting that TRRAP is a transcriptional repressor of these genes.

To strengthen this observation, we used ChIP followed by qPCR to measure RNA polymerase II binding over the gene body of both *IRF7* and *IRF9*, upon a time course of TRRAP depletion. In parallel, we performed anti-HA CUT&RUN experiments followed by qPCR to compare changes in TRRAP and RNA polymerase II occupancy. ChIP-qPCR revealed that RNA polymerase II recruitment increases at *IRF7* and *IRF9* at all time points after auxin addition compared to control conditions (*Figure 6B*). CUT&RUN-qPCR experiments confirmed the robust and specific binding of TRRAP to the promoter regions of both *IRF7* and *IRF9* compared to control IgGs and showed that auxin induces a loss of its binding (*Figure 6C*). Notably, RNA polymerase II occupancy increases already 7 hr after auxin addition, while TRRAP is already undetectable at their promoters, suggesting that TRRAP depletion rapidly induces the transcription of both genes. Altogether, CUT&RUN profiling, nascent RNA, and kinetic analysis of RNA polymerase II occupancy indicate that TRRAP binds to the *IRF7* and *IRF9* promoters and represses their transcription. We thus conclude that, unexpectedly, TRRAP is a direct transcriptional repressor of master regulators of ISG expression in HCT116 cells.

## Dynamics of ISG regulation by TRRAP

To further support a direct role for TRRAP in repressing *IRF7* and *IRF9*, we studied the dynamics of their expression and of TRRAP promoter recruitment, taking advantage of the reversibility of the AID system (*Nishimura et al., 2009*). For this, after complete auxin-induced TRRAP depletion, ^AID^TRRAP cells were shifted to culture medium lacking auxin for various time points (*Figure 7A*). Western blot analyses confirmed that, following initial depletion, TRRAP progressively reaccumulates within a few hours after washing out auxin from the media (*Figure 7B*).

CUT&RUN followed by qPCR showed that TRRAP binding to *IRF7* and *IRF9* increases upon auxin removal (*Figure 7C*). Specifically, we observed that TRRAP binding progressively reappears at both the *IRF7* and *IRF9* promoters, in parallel to TRRAP recovery (*Figure 7B and C*). We found that, 6 hr after auxin removal, TRRAP occupancy is similar to that observed in control conditions, suggesting that TRRAP binding is highly dynamic at these promoters. We then sought to correlate TRRAP occupancy profile with its regulatory role in ISG expression. Quantitative RT-PCR of *IRF7* mRNA levels showed variable results but not reproducible differences, although we noticed a small decrease at later time points of auxin wash out (*Figure 7D*). In contrast, upon auxin removal, *IRF9* mRNA levels rapidly stop increasing and progressively decrease back to basal levels compared with a continuous auxin treatment (*Figure 7D*). Similarly, TRRAP reappearance promptly stabilizes the levels of the *OAS1* and *IFIT1* mRNAs, which are indirectly regulated by TRRAP (*Figure 5F*), whereas their expression continues to increase when TRRAP depletion is sustained (*Figure 7D*).

In summary, re-expressing TRRAP leads to a rapid reversion of the ISG induction phenotype. We note that the dynamics of *IRF7* and *IRF9* regulation shows slightly different profiles and suggests a more important role of TRRAP on *IRF9*. Taken together, our kinetic analyses of TRRAP binding and regulatory roles indicate that TRRAP and ISG levels dynamically anticorrelate in HCT116 cells and provide additional evidence for a direct repressive role for TRRAP in *IRF9* expression.

## Discussion

Our work defines the contribution of the HSP90 cochaperone TTT to TRRAP functions in human cells. Using an inducible degron for rapid and robust depletion of endogenous proteins, we demonstrate that the TTT subunit TELO2 promotes TRRAP assembly into functional complexes, SAGA and TIP60,

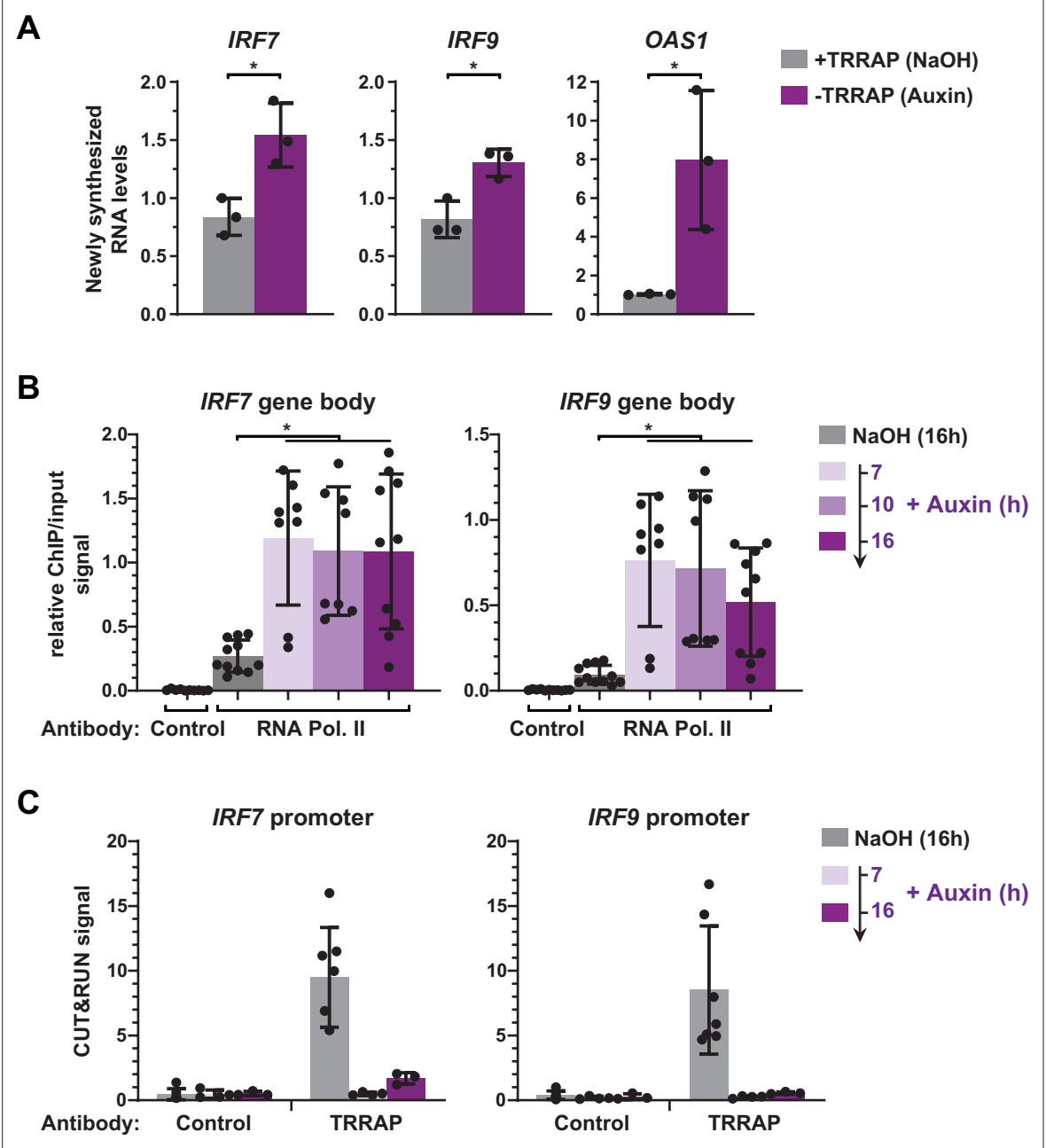

**Figure 6.** TRRAP directly represses the transcription of *IRF7* and *IRF9*. (**A**) RT-qPCR analysis of newly synthesized *IRF7*, *IRF9*, and *OAS1*. [AID]TRRAP cells were treated with either NaOH (gray) or auxin (purple) for 10 hr, prior to a 20 min incubation with 4-thiouridine (4sU), extraction, and enrichment of labeled nascent RNAs. Each value represents mean pre-mRNA levels from three independent experiments with distinct [AID]TRRAP clones, overlaid with individual data points and error bars showing the SD. *PPIB* served as a control for normalization across samples. For all genes, primers amplifying intronic regions were used. Values from one untreated replicate were set to 1, allowing comparisons across culture conditions and replicates. Statistical significance was determined by unpaired, two-tailed Student's *t*-tests. *p≤0.05. (**B**) Kinetic ChIP-qPCR analysis of RNA polymerase II occupancy at the *IRF7* and *IRF9* gene bodies. [AID]TRRAP cells were treated with NaOH for 16 hr and auxin for 7, 10, and 16 hr, as indicated, prior to chromatin extraction, sonication, and immunoprecipitation with either control rabbit IgGs or an antibody recognizing total RNA polymerase II (F-12). Each value represents mean IP/input ratios from at least eight independent experiments, overlaid with individual data points and error bars showing the SD. Statistical significance was determined by one-way ANOVA followed by Tukey's multiple comparison tests. *p≤0.05. (**C**) CUT&RUN-qPCR analysis of TRRAP occupancy at *IRF7* and *IRF9* promoters. [AID]TRRAP cells were treated with either NaOH for 16 hr or auxin for 7 and 16 hr, as indicated, prior to CUT&RUN-qPCR experiments performed using either control rabbit IgGs or an anti-HA antibody, to target endogenous TRRAP. Each column represents the mean footprint signal measured from at least three independent experiments, overlaid with individual data points and error bars showing the SD.

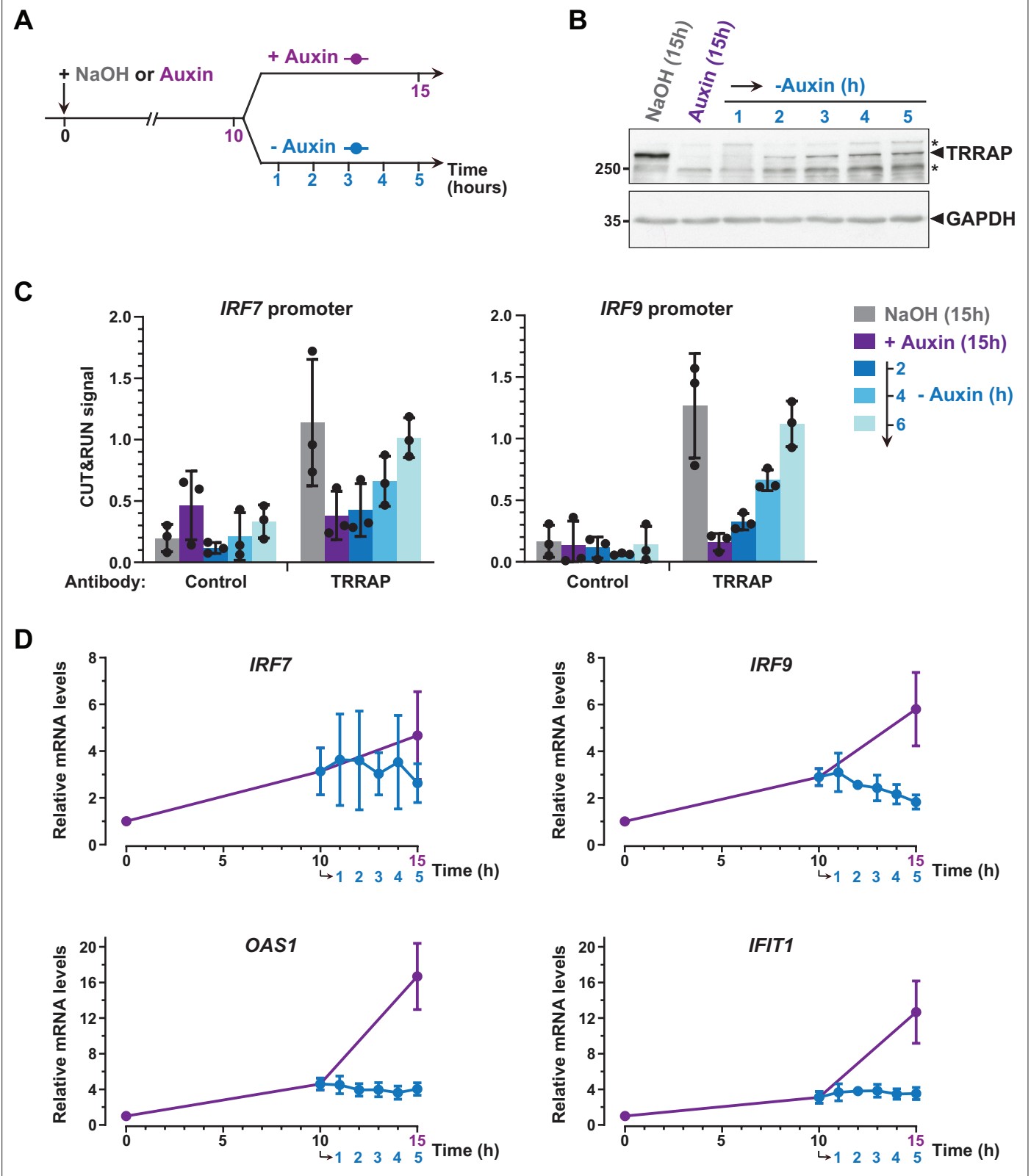

**Figure 7.** Dynamics of interferon-stimulated gene (ISG) regulation by TRRAP. (**A**) Experimental strategy: [AID]TRRAP cells were harvested before auxin addition, after incubation with auxin for 10 hr (purple), and at various time points up to 5 hr after auxin removal (blue), as indicated. As a control, [AID]TRRAP cells were also harvested following 15 hr of either NaOH (gray) or auxin (purple) treatment. (**B**) Western blot analysis of TRRAP protein levels after auxin addition and subsequent removal, demonstrating the reversibility of TRRAP auxin-mediated degradation. Blots were probed with anti-TRRAP

*Figure 7 continued on next page*

*Figure 7 continued*

and anti-GAPDH antibodies. The latter was used to control for equal loading. * marks nonspecific bands detected by the anti-TRRAP antibody. Source data are available in supplementary material (***Figure 6B***, ***Source data 1***). (**C**) CUT&RUN-qPCR analysis of TRRAP-bound DNA extracted following micrococcal nuclease (MNase) cleavage directed either by an anti-HA antibody (TRRAP) or control IgG (control). CUT&RUN was performed in ᴬᴵᴰTRRAP cells treated with either NaOH (gray) or auxin (purple) for 15 hr as controls. In parallel, ᴬᴵᴰTRRAP cells were also treated with auxin for 10 hr and then washed out of auxin for 2, 4, and 6 hr (blue). qPCR was performed using oligonucleotides that amplify proximal regions of the *IRF7* and *IRF9* promoters, as in ***Figure 6C***. Each column represents the mean footprint signal measured from three distinct ᴬᴵᴰTRRAP clones, overlaid with individual data points and error bars showing the SD. (**D**) RT-qPCR analysis of *IRF7*, *IRF9*, *OAS1*, and *IFIT1* mRNA levels after auxin addition and subsequent removal. RNAs were extracted from cells treated as schematized in (**A**). Each point represents the mean value of three distinct ᴬᴵᴰTRRAP clones, overlaid with error bars showing the SD. *PPIB* served as a control for normalization across samples. Values from untreated samples were set to 1, allowing comparisons across culture conditions and replicates.

and contributes to TRRAP regulatory roles in gene expression. Functional and transcriptomic analyses indicate that both TELO2 and TRRAP promote proliferation of CRC cells and regulate an overlapping set of genes, including MYC targets. We also uncovered an unexpected negative role of TRRAP in the expression of a subset of type I ISGs. Notably, we accumulated evidence that TRRAP acts as a transcriptional repressor of *IRF9*, which encodes a subunit of the ISGF3 regulatory complex. Our findings suggest that TRRAP and its chaperone TTT might contribute to colorectal tumorigenesis by promoting a specific transcriptional program, both as a coactivator of c-MYC and a repressor of specific ISGF3 targets.

## Chaperone-mediated biogenesis and regulation of the TRRAP pseudokinase

TRRAP is a member of the PIKK family of atypical kinases. Studies in mammals revealed that the pleiotropic HSP90 chaperone is specifically recruited to PIKKs by a dedicated cochaperone, the TTT complex, to promote their stability and incorporation into active complexes (***Takai et al., 2007***; ***Anderson et al., 2008***; ***Takai et al., 2010***; ***Hurov et al., 2010***; ***Kaizuka et al., 2010***; ***Izumi et al., 2012***). In contrast, the effect of TTT on the TRRAP pseudokinase, the only inactive PIKK, is less characterized. Previous work showed that TTT stabilizes TRRAP in human cells (***Takai et al., 2007***; ***Hurov et al., 2010***; ***Kaizuka et al., 2010***; ***Izumi et al., 2012***). Furthermore, the yeast ortholog of TRRAP, Tra1, also requires TTT and Hsp90 for stability, incorporation into the NuA4 and SAGA coactivator complexes, and function in gene expression (***Elías-Villalobos et al., 2019b***; ***Genereaux et al., 2012***). Here, conditional and rapid depletion of endogenous TELO2 allowed us to better characterize the role of TTT in TRRAP biogenesis and functions in human cells. Our results suggest a model by which TELO2 promotes the cytoplasmic assembly of TRRAP-containing complexes, TIP60 and SAGA, and regulates the expression of a large number of TRRAP-dependent genes. Altogether, these studies indicate that TTT functional roles are conserved between yeast and mammals. Therefore, despite its lack of catalytic activity, Tra1/TRRAP shares a dedicated chaperone machinery with active PIKK kinases for assembly and function. Phylogenetic analyses of PIKK orthologs show that Tra1/TRRAP appeared early in the eukaryotic lineage, concomitantly with all other PIKKs (***Elías-Villalobos et al., 2019a***). Although catalytic residues diverge, all TRRAP orthologs harbor the distinctive domain architecture of PIKKs, including the highly conserved FAT and kinase domains. Similar to PIKKs, orthologs of TELO2 and TTI1 are found in the genomes of species representative of all major eukaryotic clades (***Elías-Villalobos et al., 2019a***). We thus propose that the requirement of PIKKs for a dedicated cochaperone explains the selection pressure observed on the sequence and domain organization of the Tra1/TRRAP pseudokinase.

Finally, our work indicates that assembly of the NuA4/TIP60 and SAGA complexes requires the HSP90 cochaperone TTT, contributing to the emerging concept that dedicated mechanisms and chaperones control the de novo assembly of transcription complexes. For example, a recent study showed that human TAF5 and its paralog TAF5L, which are specific to the general transcription factor TFIID and SAGA, respectively, require the CCT chaperonin for incorporation into preassembled modules (***Antonova et al., 2018***). An alternative mechanism, involving cotranslational interactions between other SAGA subunits, was recently described in yeast and human cells (***Kassem et al., 2017***; ***Kamenova et al., 2019***).

## Inhibitory roles of TRRAP in ISG expression

We show here that TELO2 and TRRAP regulate the expression of an overlapping set of genes. Notably, we found that TELO2 is important for the activation of MYC and E2F target genes (*Figure 2*). In agreement, TRRAP is essential for transcription activation by MYC and E2F and contributes to their oncogenic functions during tumorigenesis (*McMahon et al., 1998*; *Park et al., 2001*; *Bouchard et al., 2001*; *Lang et al., 2001*; *Nikiforov et al., 2002*). Unexpectedly, we found that TELO2 and TRRAP also repress several genes, particularly those mediating the IFN-I response during innate immunity. Because TRRAP has well-established roles in transcription activation, rather than repression, we decided to characterize this phenotype further.

The IFN-I signaling pathway is divided into two branches that define an early and a late response, and involves five major transcription factors. Following detection of 'non-self' elements, such as pathogens, PRR signaling leads to activation of the IRF3 and IRF7 transcription factors and production of IFN-Is during the early response. Then, these cytokines activate autocrine and paracrine JAK kinase signaling to induce the formation of the ISGF3 transcription factor complex during the late response. ISGF3 comprises the IRF9, STAT1, and STAT2 transcription factors, which activate several ISGs encoding downstream effectors of IFN-I signaling and innate immunity. ISGF3 also activates *IRF7*, *IRF9*, *STAT1*, and *STAT2* expression to establish a positive feedback loop (*Figure 3A*). We found that neither the PRR nor the IFN-I signaling pathways are activated upon TRRAP depletion. Rather, TRRAP represses a specific subset of ISGs, named U-ISGs, likely by directly inhibiting *IRF7* and *IRF9* transcription. Four lines of evidence support this model. First, TRRAP depletion induces ISGs independently of PRR and IFN-I signaling pathway activation, as shown by the lack of IRF3 and STAT1 phosphorylation, respectively (*Figure 3B*). Second, time-course analyses suggest that IRF7 and IRF9 accumulation precedes that of its target genes (*Figure 3C–E*). Third, TRRAP specifically represses U-ISGs, which can be induced by the sole overexpression of unphosphorylated IRF9, including in CRCs (*Kolosenko et al., 2015*; *Cheon et al., 2013*; *Sung et al., 2015*; *Platanitis et al., 2019*; *Blaszczyk et al., 2015*; *Figure 4*). Fourth, TRRAP dynamically binds to the promoters of *IRF7* and *IRF9* and represses their transcription (*Figures 5–7*). Although both IRF7 and IRF9 likely contribute to ISG induction upon TRRAP depletion, we propose that IRF9 mediates the effect of TRRAP on U-ISGs expression. Indeed, ISGF3, particularly IRF9, is the critical effector of IFN-I signaling and induces the expression of ISGs downstream of TBK1 and IRF7 activation. In addition, IRF9 can activate *IRF7* expression itself, including in HCT116 cells (*Kolosenko et al., 2015*). Altogether, the rapid kinetics, efficiency, and reversibility of the AID allowed us to deduce a putative sequence of phenotypic events, by which TRRAP depletion would relieve *IRF9* repression, subsequently inducing the transcription of *IRF7* and a subset of other U-ISGs, independently of IFN-I signaling. Nevertheless, we note that seven additional U-ISGs are both repressed and bound by TRRAP and thus putative direct targets (*Figure 5F*). These genes encode key components of the IFN-I signaling pathway and include the viral RNA sensors DDX58 (RIG-I) and IFIH1 (MDA5), the signaling adaptor IFIT3, and the ISGF3 transcription complex subunit STAT1. Although we did not test this possibility, TRRAP might directly contribute to their repression, reinforcing its inhibitory effect on *IRF7* and *IRF9* expression.

An alternative mechanism is that TRRAP activates the transcription of a yet uncharacterized repressor of ISGs, particularly of IRF9. Although we cannot formally exclude this possibility, examination of the list of genes bound and activated by TRRAP did not identify an obvious candidate. However, so far, little is known about the regulation of *IRF9* expression, both in immune cells and cancer tissues (*Suprunenko and Hofer, 2016*). Gene expression databases indicate that *IRF9* is expressed constitutively. Accordingly, previous studies demonstrated that its regulation primarily occurs at the level of ISGF3 complex formation and nucleocytoplasmic shuttling (*Wang et al., 2017*). ISGF3 activation does not require de novo gene expression, although IFN-I signaling induces expression of *IRF9* and other transcription actors in a positive feedback loop. Further work is therefore needed to identify direct regulators of *IRF9* transcription, particularly repressors, to test whether TRRAP can also inhibit IRF9 expression indirectly.

How does TRRAP represses specific ISGs? Several studies suggest that both the TIP60 and SAGA complexes might be involved. A proximity labeling screen identified EP400, which catalyzes H2A.Z deposition, as a putative interactor of IRF9 (*Platanitis et al., 2019*). Furthermore, H2A.Z is removed during ISGF3-dependent transcription activation and, conversely, H2A.Z silencing induces ISGF3 recruitment to ISG promoters, ISG mRNA expression, and IFN-stimulated antiviral immunity

(*Au-Yeung and Horvath, 2018*). However, we did not detect a reproducible effect of TRRAP on H2A.Z occupancy at the *IRF9* promoter (our unpublished observations). Another study identified TIP60 as a repressor of endogenous retroviral elements (ERVs), which can activate innate immune signaling pathways when expressed (*Rajagopalan et al., 2018*; *Chiappinelli et al., 2015*; *Roulois et al., 2015*). However, we did not detect activation of an innate immune response in TRRAP-depleted cells. Moreover, these studies show that TIP60 depletion does not induce expression of IRF9 and its target ISGs, suggesting a different mechanism. In contrast, recent work in *Drosophila* found that the Tip60 subunit E(Pc) attenuates expression of components of JAK/STAT signaling (*Bailetti et al., 2019*), which is reminiscent of our findings and suggests that this function of TRRAP might be conserved in other eukaryotic species. Finally, genetic and biochemical evidence indicates that the SAGA subunit GCN5/PCAF represses IFN-I signaling by inhibiting the TBK1 kinase, which targets IRF3 (*Jin et al., 2014*). In contrast, we observed that TRRAP depletion induces ISG expression without IRF3 phosphorylation and activation of IFN signaling, demonstrating that TRRAP functions by a distinct mechanism in the context of CRC HCT116 cells.

### ISG expression in colorectal cancer
Alterations of TIP60 and SAGA activities have been documented in various cancers, for example, as essential cofactors of pro-oncogenic transcription factors (*Wang and Dent, 2014*; *Judes et al., 2015*). The shared subunit TRRAP represents a compelling example because it was identified in a screen for factors that are essential for MYC-dependent transcription and malignant transformation (*McMahon et al., 1998*). Supporting this conclusion, TRRAP is recurrently overexpressed in CRC patient samples (our analysis of the COADREAD cohort, http://xena.ucsc.edu/), and we show here that TRRAP depletion primarily affects a MYC target signature in HCT116 cells. In addition, our work suggests that TRRAP might also contribute to colorectal tumorigenesis by restricting ISG expression levels. Indeed, several studies reported that ISGs are normally repressed in CRC cells, but upregulated upon acquisition of radio- and chemoresistance (*Cheon et al., 2013*; *Gongora et al., 2008*; *Khodarev et al., 2004*; *Weichselbaum et al., 2008*; *Khodarev et al., 2007*; *Fryknäs et al., 2007*). Specifically, gene expression profiling identified a subset of ISGs, defined as the interferon-related DNA damage signature (IRDS), which increases in cancer cells that become resistant to ionizing radiation, topoisomerase inhibitors, or DNA damage. Remarkably, the IRDS is virtually identical to the subset of ISGs induced by unphosphorylated U-ISGF3 and those repressed by TRRAP in HCT116 CRC cells. Therefore, TRRAP might play a new role in colorectal tumorigenesis by repressing a subset of ISGs.

## Materials and methods
### Cell culture, reagents, and transfection
HCT116 cells (a gift from Dr. Vogelstein, Howard Hughes Medical Institute, Baltimore, MD) were cultured in McCoy's 5A medium (Sigma) supplemented with 10% (v/v) FBS and 100 U/ml penicillin/streptomycin in 5% $CO_2$ at 37°C. Cells were regularly tested negative for mycoplasma. Cell line authentication was performed by Eurofins using a standardized STR profiling procedure and the Applied Biosystems AmpFLSTRTM Identifiler Plus PCR Amplification Kit. For siRNA knockdown, HCT116 cells were seeded in 6-well plates 24 hr before transfection and transfected with 20 nM of siRNA using INTERFERin (Polyplus Transfections) according to the manufacturer's protocol. The siRNA sequences are GCAACAAGCUUUAGAACUA and UGAAGUAGAUGUGGAGAAA for targeting *SUPT20H*, and CUUACGCUGAGUACUUCGA for targeting firefly luciferase. Cells were then harvested 48 hr after transfection for RNA isolation and protein analysis. For auxin treatment, 3-indoleacetic acid (IAA) (Sigma, I2886) was dissolved in NaOH 1 N and used at a final concentration of 0.5 mM. For interferon pathway activation, 2 µg of polyinosinic-polycytidylic acid (poly(I:C) LMW) (InvivoGen, tlrl-picw) were transfected. For cell proliferation assays, cells were seeded at 20,000 cells per well in 1 ml in 24-well plates. Cells were counted daily using the Countess automated cell counter based on trypan blue exclusion assay (Invitrogen).

### Generation of CRISPR-Cas9-edited cell lines
Single-guide RNAs (sgRNAs) were designed using the http://crispr.mit.edu/ website. Then, candidate sgRNAs with the highest scores (indicating fewest potential off-targets) were selected and

synthesized. Two complementary oligonucleotides of sgRNAs were annealed and cloned into the BbsI sites of pUC57 vector (*Supplementary file 5*). The two different sgRNA plasmids, the pX335-U6-Chimeric_BB-CBh-hSpCas9n(D10A) plasmid (Addgene, #42335) and the donor plasmid containing the AID and YFP cassettes, were co-transfected into an HCT116 cell line stably expressing *Os*TIR1-9Myc using the FuGENE6 transfection reagent (Promega). YFP-positive cells were selected and isolated by fluorescence-activated cell sorting 48 hr after transfection.

## Cell fractionation

Separation of cytoplasmic and nuclear fractions was achieved using the Rapid, Efficient and Practical (REAP) protocol (*Suzuki et al., 2010*). Briefly, culture medium was removed from cell culture dishes and cells were washed twice using ice-cold PBS. 1 ml of PBS was added to each 10 cm dish before cells were scraped and collected in microcentrifuge tubes. Samples were centrifuged (10 s at 2000 × *g*) and supernatants were discarded. 1 ml of 0.1% NP40-PBS1x was added to each pellet with pipetting up and down several times. Small aliquots (150 µl) were set aside as whole-cell lysates (WCE). The rest of the samples were centrifuged and supernatants were transferred to new tubes, constituting the cytoplasmic fractions (150 µl) (CE). Pellets were resuspended in 1 ml 0.1% NP40 and centrifuged to obtain nuclear fractions as pellets (NE). Whole lysates were mixed with 4× Laemmli buffer (ratio 3:1), sonicated, and boiled. Cytoplasmic fractions were mixed with 4× Laemmli buffer (ratio 3:1) and boiled. Nuclear fractions were resuspended in 200 µl of 1× Laemmli buffer, sonicated, and boiled. For Western blot analyses, 20 µl of WCE and CE fractions and 10 µl of NE were loaded.

## Cell lysis, Immunoprecipitation, and Western blots

Cells from 6-well plates were harvested by trypsinization and lysed in RIPA buffer 20 mM Tris-HCl pH 7.5, 150 mM NaCl, 1% Nonidet-P40, 0.5% sodium deoxycholate, protease inhibitors (cOmplete EDTA-free Cocktails Tablets, Roche). For immunoprecipitation, 1-5 mg of cell lysates were incubated with antibody-bound protein G-Sepharose beads (GE Healthcare Biosciences, Pittsburgh, PA) overnight at 4°C. Beads were washed and resuspended in five times in 20 mM Tris-HCl pH7.5, 150 mM NaCl, 0.1% Triton and resuspended in 1x Laemmli buffer. Cell lysates were subjected to 6 or 10% SDS-PAGE and 40 µg of protein were loaded per lane. The gels were transferred for 2 hr onto nitrocellulose membranes (GE Healthcare Biosciences, Pittsburgh, PA). Detection was performed by diluting primary antibodies in TBS supplemented with 0.1% Tween and either 5% BSA or 5% non-fat dry milk. All antibodies used in this study are listed in *Supplementary file 6*. After incubation and washing, secondary anti-mouse IgG or anti-rabbit IgG antibodies conjugated to horse radish peroxidase (Santa Cruz) were added at a dilution of 1:5000 in TBS 0.1% Tween 5% non-fat drymilk. Detection was performed using Pierce ECL Western blotting Substrate (Thermo Fisher Scientific).

## RT-qPCR

Total RNA was isolated from HCT116 cells using TRIzol reagent (Invitrogen, 15596018), according to the manufacturer's instructions, followed by a DNAse digestion step using the TURBO DNA-free kit (Ambion, AM1907). Total RNA (1 µg) was reverse-transcribed using the SuperScript III reverse transcriptase (Invitrogen, Life Technologies). Real-time quantitative PCR (qPCR) was performed using SYBR Green Master Mix and the LightCycler 480 instrument (Roche). Relative levels of gene expression were analyzed using the 2ΔΔCt method and compared to the expression of the human housekeeping gene *PPIB*. The cycling conditions comprised an initial denaturation phase at 95°C for 5 min, followed by 50 cycles at 95°C for 10 s, 60°C for 30 s, and 72°C for 15 s. All oligonucleotides used for amplification are listed in *Supplementary file 5*.

## Nascent RT-qPCR

Metabolic labeling of newly transcribed RNAs was performed as described in *Schwalb et al., 2016*. Briefly, the nucleoside analogue 4sU (Abcam, ab143718) was added to culture medium at a final concentration of 500 µM for a 20 min pulse. 4sU was removed by washing cells with ice-cold 1× PBS and immediately lysed using TRIzol reagent (Invitrogen, 15596018). Total RNA was extracted according to the manufacturer's instruction and precipitated. For purification, fragmented total RNA was incubated for 10 min at 60°C and immediately chilled on ice for 2 min to melt secondary RNA structures. 4sU-labeled RNA was thiol-specific biotinylated by addition of 200 µg EZ-link HPDP-biotin

(Thermo Fisher Scientific, 21341), biotinylation buffer (10 mM HEPES-KOH pH 7.5 and 1 mM EDTA), and 20% DMSO (Sigma-Aldrich, D8418) to prevent precipitation of HPDP-biotin. Biotinylation was carried out for 3 hr at 24°C in the dark and with gentle agitations. After incubation, biotin excess was removed by adding an equal volume of chloroform and centrifugation at 16,000 × $g$ for 5 min at 4°C. RNA was precipitated from the aqueous phase using 0.1 volumes of 5 M NaCl and one volume of 100% isopropanol followed by centrifugation at 16,000 × $g$ for 30 min at 4°C. After washing with 75% ethanol, the RNA pellet was resuspended in 100 µl of RNase-free water and denatured for 10 min at 65°C followed by immediate chilling on ice for 5 min. RNA samples were incubated with 100 µl of streptavidin-coated µMACS magnetic beads (Miltenyi Biotec, 130-074-101) for 90 min at 24°C under gentle agitations. The µMACS columns (Miltenyi Biotec, Cat# 130-074-101) were placed on a MACS MultiStand (Miltenyi Biotec) and equilibrated with washing buffer (100 mM Tris-HCl pH 7.5, 10 mM EDTA, 1 M NaCl, 0.1% Tween20) before applying the samples twice to the columns. The columns were then washed one time with 600 µl, 700 µl, 800 µl, 900 µl, and 1 ml washing buffer before eluting the newly synthesized RNA with two washes of 100 µl 0.1 M DTT. Purified newly synthesized RNAs were recovered using the RNeasy MinElute Cleanup Kit (QIAGEN, 74204), according to the manufacturer's instruction. 1 µg RNA was reverse-transcribed using the SuperScript III reverse transcriptase (Invitrogen, Life Technologies). RT-qPCR was performed using SYBR Green Master Mix and the LightCycler 480 instrument (Roche). Relative levels of gene expression were analyzed using the 2ΔΔCt method and compared to the expression of the human housekeeping gene *PPIB*. The cycling conditions comprised an initial denaturation phase at 95°C for 5 min, followed by 50 cycles at 95°C for 10 s, 60°C for 30 s, and 72°C for 15 s. All oligonucleotides used for amplification are listed in *Supplementary file 5*.

## Chromatin immunoprecipitation

Following NaOH or IAA treatment, adherent cells were fixed with 1% paraformaldehyde for 8 min at room temperature, which was subsequently quenched with 125 mM glycine. After washing with PBS and scrapping cells, nuclear extracts were prepared in a two-step lysis-centrifugation procedure using, first, a hypotonic buffer (50 mM Tris-HCl pH 8.0, 100 mM NaCl, 5 mM MgCl$_2$, 0.5% Nonidet-P40, protease inhibitors [cOmplete EDTA-free Cocktails Tablets, Roche]) and, second, an SDS-lysis buffer (50 mM Tris-HCl pH 8.0, 10 mM EDTA pH 8.0, 1% SDS, protease inhibitors). Extracts were sonicated for 10 min in 30 s ON/30 s OFF cycles using a Bioruptor Pico sonicator (Diagenode). Sonicated extracts were then diluted 7.5× in immunoprecipitation dilution buffer (50 mM Tris-HCl pH 8.0, 167 mM NaCl, 1 mM EDTA pH 8.0, 1.1% Triton X-100, 0.01% SDS, protease inhibitors) and incubated with 2 µg of either IgG control or anti-Pol II antibodies (F-12) overnight at 4°C and then with Dynabeads Protein G (Thermo Fisher Scientific) for 2 hr. Beads were washed for 15 min at 4°C on a rotating wheel with, successively, low-salt buffer (50 mM Tris-HCl pH 8.0, 150 mM NaCl, 1 mM EDTA, 1% Triton X-100, 0.1% SDS), high-salt buffer (50 mM Tris-HCl pH 8.0, 500 mM NaCl, 1 mM EDTA, 1% Triton X-100, 0.1% SDS), LiCl buffer (20 mM Tris-HCl pH 8.0, 250 mM LiCl, 1 mM EDTA, 1% Nonidet-P40, 1% Na-deoxycholate), and elution buffer (10 mM Tris pH 8.0, 1 mM EDTA, 0.02% Tween-20). DNA was recovered using 200 µl of elution buffer supplemented with 100 mM NaHCO$_3$ and 1% SDS. Crosslinks were reversed by incubating samples overnight at 65°C and treated with RNAse A and proteinase K for 2 hr at 55°C. DNA was extracted using NucleoSpin columns (Macherey-Nagel) and eluted in 60 µl of NE buffer and used for qPCR analysis using SYBR Green Master Mix and the LightCycler 480 instrument (Roche).

## RNA-seq

Total RNA was extracted using the TRIzol reagent (Invitrogen, 15596018) according to the manufacturer's instructions, followed by DNAse digestion step using the TURBO DNA-free kit (Ambion, AM1907). RNA concentration and quality were analyzed using an Agilent Bioanalyzer 2100 (Agilent Technologies). Total RNA-seq libraries were generated from 1 µg of total RNA using TruSeq Stranded Total RNA LT Sample Prep Kit. High-throughput sequencing of cDNA libraries was performed following the standard protocol from Fasteris (https://www.fasteris.com). All sequencing runs were performed on an Illumina HiSeq 2500.

Transcripts were purified by polyA-tail selection. Stranded dual-indexed cDNA libraries were constructed using the Illumina TruSeq Stranded mRNA Library Prep kit. Library size distribution and

concentration were determined using an Agilent Bioanalyzer. 12 libraries were sequenced in one lane of an Illumina HiSeq 4000, with 1 × 50 bp single reads, at Fasteris SA (Plan-les-Ouates, Switzerland). After demultiplexing according to their index barcode, the total number of reads ranged from 23 to 27 million per library. Adapter sequences were trimmed from reads in the FastQ sequence files. Reads were aligned using HISAT2 (*Kim et al., 2015*), with strand-specific information (–rnastrandness R) and otherwise default options to human genome assembly GRCh37 (hg19). For all samples, the overall alignment rate was over 80%, including over 90% of reads mapping uniquely to the human genome. Reads were then counted for exon features using htseq-count (*Anders et al., 2015*) in union mode (–mode union), reverse stranded (–stranded Reverse), and a minimum alignment quality of 10 (–minaqual 10). For all samples, over 95% of reads were assigned to a feature (–type gene). Variance mean dependence was estimated from count tables and tested for differential expression based on a negative binomial distribution using DESeq2 (*Love et al., 2014*). Pairwise comparison or one-way ANOVA were run with a parametric fit and 'treatment as the source of variation' (factor: auxin or NaOH). All computational analyses were run either on R (version 3.6.0), using publicly available packages, or on the Galaxy web platform using the public server at usegalaxy.org (*Afgan et al., 2018*).

## CUT&RUN

CUT&RUN experiments were implemented based on the procedure described in *Meers et al., 2019*. Briefly, cells were grown up to 80% confluence, harvested from fresh cultures, and counted. About 250,000 cells were used per CUT&RUN sample. Cells were washed and incubated for 10 min at room temperature with 10 μl of concanavalin A-coated beads. Cells bound to beads were permeabilized and incubated with a ChIP-grade anti-HA antibody (ab9110, Abcam) at 4°C overnight. Protein A-MNAse (pA-MN) was added to a final concentration of 700 ng/ml and incubated with cells at 4°C for 1 hr. Next, pA-MN was activated with 2 μl of 100 mM $CaCl_2$ and digestion was performed for 30 min at 0°C. The reaction was stopped with 100 μl of stop buffer containing 2 pg/ml of sonicated genomic DNA from *Drosophila* S2 cells as a heterologous spike-in. Release of the DNA fragments was achieved by incubating samples at 37°C during 15 min. DNA was extracted following using NucleoSpin columns (Macherey-Nagel) and eluted in 30 μl of NE buffer. Extracted DNA was used either for qPCR analysis or for Illumina high-throughput sequencing. qPCR was performed using the SYBR Green Master Mix and the LightCycler 480 instrument (Roche). Ct values were transformed into a DNA quantity value using a sample of genomic DNA of known concentration in each qPCR plate. Data were then analyzed and graphed using GraphPad Prism without any further normalization. All oligonucleotides used for amplification are listed in *Supplementary file 5*.

For sequencing, barcoded CUT&RUN-seq libraries were constructed using the KAPA Single-Indexed Adapters set A&B and the Hyper Prep kit (KAPA Biosystems), following the manufacturer's instructions, except that pooled DNA samples were amplified for 12 cycles, with a combined annealing-extension step for 10 s at 60°C. Yield and size distribution were quantified on a 2100 Bioanalyzer instrument (Agilent). High-throughput sequencing of libraries was performed on an Illumina HiSeq 3000/4000, with 2 × 150 pb paired-end reads, at Fasteris SA (Plan-les-Ouates). After demultiplexing according to their index barcode, the total number of reads ranged from 2.7 to 4.5 million per library. Illumina adapter sequences were trimmed using Trimmomatic tool (version 0.36.5). Reads were aligned on the GRCh37 (hg19) human genome assembly using Bowtie 2 (version 2.3.4.2) (*Langmead and Salzberg, 2012*), with paired-end option fragment length for valid paired-end alignments set from 10 to 700, and otherwise default options. Peak predictions for TRRAP occupancy were performed with the Model-based Analysis for ChIP-seq MACS2 (version v2.1.1.20160309.0) (*Zhang et al., 2008*), with the following parameters: `--format` BAMPE, `--bw` 300, `--mfold` and `--qvalue` set to default values and `--bdg`. For both the NaOH and IAA treatment conditions, two independent anti-HA replicate samples were analyzed against one control, rabbit IgG sample. Read density was visualized with the Integrative Genomics Viewer (IGV) (*Robinson et al., 2011*), and heatmap profiles of TRRAP occupancy were represented using seqMINER (version 1.3.4). Heatmaps show only one replicate for each condition, without any normalization (*Ye et al., 2011*). Each MACS2 peak was associated to the nearest TSS and their cognate gene symbols using the Closest tool from the bedtools suite (https://bedtools.readthedocs.io/en/latest/). Acetyl H3K27 and trimethyl H3K4 ChIP-seq profiles from HCT116 cells were published previously (*Zhang et al., 2020*) and obtained from GEO, through accession numbers GSM945853 and GSM945304, respectively. All computational analyses were run

using publicly available packages on the Galaxy web platform 9 using the public server at usegalaxy. org (*Afgan et al., 2018*).

## Statistics

Statistical tests were performed using GraphPad Prism. *t*-tests were used when comparing two means. One-way or two-way ANOVA were performed for comparing more than two means, across one (e.g., 'genotype') or two distinct variables (e.g., 'genotype' and 'treatment'). ANOVAs were followed by Bonferroni or Tukey post hoc pairwise comparisons. Comparisons that are statistically significant (p≤0.05) are marked with *, whereas those that are statistically not significant (p>0.05) are labeled n.s.

## Acknowledgements

We thank Manon Le Goff, Aida Yazbeck, and Boutaina El Kenz for invaluable technical assistance and all members of the Helmlinger laboratory for helpful suggestions and discussions. We are grateful to Véronique Gire, Benjamin Vitre, Daniele Fachinetti, Edouard Bertrand, Didier Devys, and Laszlo Tora for kindly sharing plasmids and cell lines. We are indebted to Véronique Fischer and Steve Henikoff for sharing reagents and helping set up the nascent RNA and CUT&RUN analyses, respectively. We thank Léo Pioger, Emeric Dubois, and Samia Guendouz for their help with RNA-seq. DD is a recipient of a graduate fellowship from la Ligue Nationale Contre le Cancer. This work was supported by funds from the Fondation ARC (PJA-20131200471 and PJA-20181208277), the Ligue Contre le Cancer, Comité départemental de l'Hérault, and the INCa grant PLBIO 2016-161.

## Additional information

### Funding

| Funder | Grant reference number | Author |
|---|---|---|
| Fondation ARC pour la Recherche sur le Cancer | PJA-20181208277 | Dominique Helmlinger |
| Institut National Du Cancer | PLBIO 2016-161 | Berengere Pradet-Balade Dominique Helmlinger |
| Ligue Nationale Contre le Cancer | Graduate Student Fellowship | Dylane Detilleux |
| Ligue Nationale Contre le Cancer | Comité Hérault | Peggy Raynaud |

The funders had no role in study design, data collection and interpretation, or the decision to submit the work for publication.

### Author contributions

Dylane Detilleux, Formal analysis, Investigation, Methodology, Visualization, Writing - original draft, Writing - review and editing; Peggy Raynaud, Formal analysis, Investigation, Methodology, Writing - review and editing; Berengere Pradet-Balade, Formal analysis, Funding acquisition, Investigation, Methodology, Writing - review and editing; Dominique Helmlinger, Conceptualization, Formal analysis, Funding acquisition, Investigation, Supervision, Visualization, Writing - original draft, Writing - review and editing

### Author ORCIDs

Peggy Raynaud ![ORCID] http://orcid.org/0000-0003-1224-8495
Berengere Pradet-Balade ![ORCID] http://orcid.org/0000-0003-1720-3739
Dominique Helmlinger ![ORCID] http://orcid.org/0000-0003-1501-0423

### Decision letter and Author response

Decision letter https://doi.org/10.7554/eLife.69705.sa1
Author response https://doi.org/10.7554/eLife.69705.sa2

# Additional files

## Supplementary files

• Supplementary file 1. Gene expression changes upon TELO2 depletion. Shown are DESeq2 results from RNA-seq experiments comparing HCT116 TELO2-AID cells treated with either NaOH or auxin for 48 hr. log2FoldChange indicates the Log2 fold change of the ratio of RNA-seq counts in auxin-treated cells over NaOH-treated cells, averaged from three independent clones. padj indicates the p-values after Benjamin–Hochberg correction for multiple testing.

• Supplementary file 2. Gene expression changes upon TRRAP depletion. Shown are DESeq2 results from RNA-seq experiments comparing HCT116 AID-TRRAP cells treated with either NaOH or auxin for 24 hr. log2FoldChange indicates the Log2 fold change of the ratio of RNA-seq counts in auxin-treated cells over NaOH-treated cells, averaged from three independent clones. padj indicates the p-values after Benjamin–Hochberg correction for multiple testing.

• Supplementary file 3. Interferon-stimulated gene (ISG) expression changes upon TELO2 and TRRAP depletion. Shown are DESeq2 results as in Tables S1 and S2, filtered for IFN-I stimulated genes (113 ISGs). The list of 113 ISGs results from merging the 97 genes from the HALLMARK_ INTERFERON_ALPHA_RESPONSE (MSigDB) with all non-redundant U-ISGF3- and ISGF3-regulated genes. Source data for *Figure 4* and *Figure 2—figure supplement 3*.

• Supplementary file 4.Genome-wide occupancy profile of TRRAP in HCT116 cells. 'NarrowPeaks' output files from MACS2 peak calling analyses of duplicate anti-HA CUT&RUN-seq experiments performed in AID-TRRAP cells, treated with either NaOH or auxin (IAA) for 12 hr, compared to a control IgG CUT&RUN-seq sample.

• Supplementary file 5. List of oligonucleotides used in this study.

• Supplementary file 6. List of antibodies used in this study.

• Transparent reporting form

• Source data 1. This contains all original uncropped scans of all Western blots.

## Data availability

The raw sequencing data reported in this publication have been deposited in NCBI Gene Expression Omnibus and are accessible through GEO Series accession number GSE171454 and GSE192527.

The following datasets were generated:

| Author(s) | Year | Dataset title | Dataset URL | Database and Identifier |
|-----------|------|---------------|-------------|-------------------------|
| Detilleux D, Raynaud P, Helmlinger D | 2021 | The TRRAP transcription cofactor represses interferon-stimulated genes in colorectal cancer cells | https://www.ncbi.nlm.nih.gov/geo/query/acc.cgi?acc=GSE171454 | NCBI Gene Expression Omnibus, GSE171454 |
| Detilleux D, Raynaud P, Helmlinger D | 2021 | The TRRAP transcription cofactor represses interferon-stimulated genes in colorectal cancer cells | https://www.ncbi.nlm.nih.gov/geo/query/acc.cgi?acc=GSE192527 | NCBI Gene Expression Omnibus, GSE192527 |

The following previously published datasets were used:

| Author(s) | Year | Dataset title | Dataset URL | Database and Identifier |
|-----------|------|---------------|-------------|-------------------------|
| Cheon H, Stark GR | 2013 | Genes induced by IFN-beta, IFN-gamma or unphosphorylated STAT1 in human fibroblasts | https://www.ncbi.nlm.nih.gov/geo/query/acc.cgi?acc=GSE50954 | NCBI Gene Expression Omnibus, GSE50954 |
| Farnham - University of Southern California | 2012 | USC_ChipSeq_HCT-116_H3K27ac_UCDavis | https://www.ncbi.nlm.nih.gov/geo/query/acc.cgi?acc=GSM945853 | NCBI Gene Expression Omnibus, GSM945853 |
| Stamatoyannopoulous - University of Washington | 2012 | UW_ChipSeq_HCT-116_H3K4me3 | https://www.ncbi.nlm.nih.gov/geo/query/acc.cgi?acc=GSM945304 | NCBI Gene Expression Omnibus, GSM945304 |

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
