## [Editor Report]

This work will be of wide interest to the transcription community as it is the first evidence that the cofactor, TRRAP, which is known as a transcriptional activator, can also act as a transcriptional repressor. The new experiments added to the revised manuscript further support this conclusion.

---

## [Decision Letter]

**Decision letter after peer review:**

Thank you for submitting your article "The TRRAP transcription cofactor represses interferon-stimulated genes in colorectal cancer cells" for consideration by *eLife*. Your article has been reviewed by 2 peer reviewers, including Jerry L Workman as the Reviewing Editor and Reviewer #1, and the evaluation has been overseen by Kevin Struhl as the Senior Editor.

The manuscript by Detilleux and coworkers continues studies by the Helmlinger group on the conserved TRAPP subunit of the SAGA and TIP60 (NuA4 in yeast) complexes. In this manuscript the authors create auxin-inducible degron allelles of TRAPP and of its TTT chaperone TELO2 in the HCT116 colorectal cancer cell line. In previous work the group showed that the TTT complex stabilizes TRAPP, and indeed degradation of TELO2 reduces nuclear accumulation of TRAPP. As expected direct auxin-induced degradation of TRAPP is more rapid, and inhibits HTC116 cell growth already after one day instead of two days for TELO2. Loss of TRAPP from the SAGA and TIP60 is significant, but not complete. RNAseq analysis showed that reduced TELO2 and TRAPP mostly leads to a reduced expression of genes, including MYC and E2F target genes, which is expected given the documented role of TRAPP as a co-activator for MYC ad E2F.

Unexpected is the increase expression of the interferon type 1 group of genes (ISGs). The authors investigated regulation of the ISG pathway to find that TRAPP depletion mostly affects IRF9 expression at the mRNA and protein levels and to a lesser extent IRF7 expression. IRF9 and IRF7 are critical transcription factors for the ISG pathway and these observations offer an explanation for the induction of interferon type 1 genes after TRAPP depletion. The authors continue to show by 4SU labeling that IRF9 and IRF7 are transcriptionally induced and by CUT&RUN-PCR that TRAPP binds to the promoter regions of these genes. Re-expression of TRAPP reverses these effects. In order to dissect which of the TRAPP containing complexes several SAGA and TIP60 complexes are targeted by siRNA knock-down, but this does not provide clear distinction between SAGA and TIP60. In general, the exact mechanistic details of TRAPP-mediated repression of gene transcription have not been worked out, but the current work provides a strong basis for future studies addressing this.

Major open issues:

1) The reviewers did not think the authors need a detailed understanding of the repression mechanism. However, the basic conclusion has to be correct, and the various models of "indirect effect" such as TRRAP activating a repressor (which are very plausible and even more likely) are inconsistent with the main conclusion. The interest of the paper is that, although TRRAP is well described as a positive factor, it has a direct negative effect on a set of genes. However, the "indirect" models involve TRRAP function in the usual positive manner, which isn't novel or even interesting. Indirect effects happen all the time and are rarely of sufficient interest for a journal like *eLife* (and this paper is not an exception to this). More generally, reducing the function of a chromatin-modifying activity usually leads to both up- and down-regulation of many genes. So finding genes that behave in the opposite fashion from what is expected happens all the time. Without demonstrating a direct negative effect, we don't see the advance here.

One experiment would greatly help show a direct negative effect. A kinetic experiment where one rapidly deplete TRRAP and simultaneously assays TRRAP association (ChIP) and transcription (Pol II occupancy or better yet (PRO-seq or Net-seq) over multiple time points. For a direct effect, loss of TRRAP occupancy should be concomitant with increased transcription).

A good approach to this would be a time course genome wide CUT and RUN coupled with PRO-seq or Net-seq

2) It is unclear why the authors did not choose to sequence the DNA from the TRAP CUT&RUN experiment, but rather performed (a more cumbersome) PCR analysis. A genome-wide CUT&RUN dataset for TRAPP would have allowed a direct comparison with their TELO2 and TRAPP depletion RNAseq datasets.

3) The experiments implicating both TIP60 and SAGA in the repression of the IRF9 gene are not convincing. This part should be removed from the manuscript as substantial additional work would be required to make this claim convincing. To argue for a direct affect ChIP/PCR experiments on IRF9 are required. Interpretation of the expression changes observed on knock down of SAGA and TIP60 components are complicated by different efficiencies of the knockdowns and by the fact that these components can be components of other complexes and/or function independently in sub-complexes. Finally, different genes require different parts of SAGA for their expression, thus it is likely that different subunits would have different affects on any repression mechanism

4) The authors mention in the methods section that heterologous DNA was used to normalize CUT&RUN experiments but make no reference to this normalization in their figures or explained in the methods. In the presented data it is certainly not explained what "AU" (occupancy levels) corresponds to technically, while IgG controls are seemingly not used as reference point. The numbers presented are extremely variable and it is difficult to judge relative TRRAP binding to the 3 different promoters. If the CUTnRUN works so well, why not performing NGS and get a global view of TRRAP binding on the genome?

5) In addition, a drop in MYC occupancy at MIR17-HG promoter following auxin induction is observed and the authors explain this by a role for TRRAP in stabilizing MYC at its target genes. However, non-specific effects of auxin on CUT&RUN results are not ruled out. Profiling an additional factor that should not be affected by TRRAP depletion would be necessary to validate and increase confidence in the results obtained in Figure 6, where the authors look at the dynamics of ISG regulation by TRRAP over a time course after removing auxin by coupling CUT&RUN to RT-PCR analyses, and to confirm that TRRAP indeed is a direct repressor of IRF9.

*Reviewer #1 Recommendations for the authors:*

The experiments implicating both TIP60 and SAGA in the repression of the IRF9 gene are not convincing. This part should be removed from the manuscript as substantial additional work would be required to make this claim convincing. To argue for a direct affect ChIP/PCR experiments on IRF9 are required. Interpretation of the expression changes observed on knock down of SAGA and TIP60 components are complicated by different efficiencies of the knockdowns and by the fact that these components can be components of other complexes and/or function independently in sub-complexes. Finally, different genes require different parts of SAGA for their expression, thus it is likely that different subunits would have different affects on any repression mechanism

*Reviewer #2 Recommendations for the authors:*

An important recommendation on current manuscript is the inclusion of library sequence TRAPP CUT&RUN fragments rather than performing a CUT&RUN-qPCR.

---

## [Author Response]

Major open issues:1) The reviewers did not think the authors need a detailed understanding of the repression mechanism. However, the basic conclusion has to be correct, and the various models of "indirect effect" such as TRRAP activating a repressor (which are very plausible and even more likely) are inconsistent with the main conclusion. The interest of the paper is that, although TRRAP is well described as a positive factor, it has a direct negative effect on a set of genes. However, the "indirect" models involve TRRAP function in the usual positive manner, which isn't novel or even interesting. Indirect effects happen all the time and are rarely of sufficient interest for a journal like eLife (and this paper is not an exception to this). More generally, reducing the function of a chromatin-modifying activity usually leads to both up- and down-regulation of many genes. So finding genes that behave in the opposite fashion from what is expected happens all the time. Without demonstrating a direct negative effect, we don't see the advance here.One experiment would greatly help show a direct negative effect. A kinetic experiment where one rapidly deplete TRRAP and simultaneously assays TRRAP association (ChIP) and transcription (Pol II occupancy or better yet (PRO-seq or Net-seq) over multiple time points. For a direct effect, loss of TRRAP occupancy should be concomitant with increased transcription).A good approach to this would be a time course genome wide CUT and RUN coupled with PRO-seq or Net-seq

We would like to thank the reviewers for their suggestions to improve the manuscript and to gather additional evidence supporting a direct repressive role of TRRAP at these genes. As suggested, we simultaneously followed RNA Polymerase II (RNAPII) occupancy within the gene bodies of *IRF7* and *IRF9*, using ChIP, and TRRAP binding at their promoters, using CUT&RUN, over a time-course of auxin-induced depletion of TRRAP. Kinetic experiments showed that ISG expression starts to increase shortly after TRRAP is depleted (Figure 1B and 3E), between 5 and 10 hours after adding auxin. We therefore measured RNAPII and TRRAP binding at early (7 hours), middle (10 hours), and late (16 hours) time points of auxin treatment. RNAPII ChIP and TRRAP CUT&RUN experiments are presented in Figure 6B and 6C, respectively, and described in a novel paragraph from the Results section (“*TRRAP represses IRF7 and IRF9 transcription*”, page 20-21). Briefly, we observed an increase in RNAPII occupancy at both genes, using ChIP-qPCR with an antibody that recognizes all forms of RNAPII (F-12). Compared to control, NaOH-treated cells, RNAPII levels are higher already 7 hours after adding auxin and do not further increase. In parallel, we observed background TRRAP binding at the promoter regions of both genes already 7 hours after auxin addition.

This kinetic analysis supports a direct inhibitory role of TRRAP on the transcription of these genes in two ways. First, TRRAP disappearance from the promoter is concomitant with the appearance of RNAPII in the transcribed region. Second, this increase in RNAPII occupancy occurs at an early time-point of TRRAP depletion, 7 hours after adding auxin, which is about 2-3 hours after TRRAP levels become undetectable by Western blot. Moreover, the transcriptional effect of TRRAP is now supported by orthogonal evidence from nascent RT-qPCR and RNAPII ChIP experiments, shown in Figure 6A and 6B, respectively. We revised the corresponding paragraphs of the Discussion to incorporate these observations (page 26-27).

Finally, we agree that TRRAP might also repress ISGs indirectly, for example by activating a repressor of IRF7 and IRF9 expression, perhaps even redundantly with its direct regulatory role. To the best of our knowledge, no specific transcriptional repressor has been identified for *IRF9*, which regulation primarily occurs post-translationally upon activation of innate immune signaling pathways. Furthermore, examination of the list of genes regulated by TRRAP did not identify obvious candidate or factors involved in global transcriptional repression. We have amended the Abstract (page 2) and the Discussion (page 27-28) to include these considerations.

2) It is unclear why the authors did not choose to sequence the DNA from the TRAP CUT&RUN experiment, but rather performed (a more cumbersome) PCR analysis. A genome-wide CUT&RUN dataset for TRAPP would have allowed a direct comparison with their TELO2 and TRAPP depletion RNAseq datasets.

The CUT&RUN-qPCR analysis of TRRAP binding was indeed promising and we thus proceeded with Illumina high-throughput sequencing. Specifically, we performed anti-HA CUT&RUN-seq of ^AID^TRRAP cells treated with either NaOH or auxin for 12 hours, each in two independent replicates. We included a rabbit IgG sample to control for background pA-MNase activity. We processed and analyzed the data using conventional ChIP-seq bioinformatic tools, such as MACS2 for peak calling, here ran against the control IgG sample. These data are presented in a new Figure 5 and described in the Results under the “*Genome-wide profiling of TRRAP occupancy*” and “*TRRAP binds to the IRF7 and IRF9 promoters*” sections, page 18-20. We also revised the Discussion to integrate these results (page 27-28).

In summary, CUT&RUN-seq was remarkably efficient to measure TRRAP occupancy profile genome-wide and we thank the reviewers for prompting us to go on with this experiment. Notably, we confirmed the robust binding of TRRAP at the *IRF7* and *IRF9* promoters and identified 7 additional U-ISGs that are both bound and repressed by TRRAP and thus putative direct targets (Figure 5E-F).

Furthermore, comparing our CUT&RUN-seq and RNA-seq data showed a strong overlap between TRRAP binding and effect on gene expression, with about 75% of deregulated genes bound by TRRAP. This observation is important because this overlap is much higher than that obtained in a previous study comparing TRRAP binding and regulatory effects using ChIP-seq and RNA-seq, respectively (PMID: 29588376), in which only about 26% of TRRAP-regulated genes are bound by TRRAP. We conclude that CUT&RUN-seq represents a significant methodological improvement for profiling the genome-wide binding of factors that, similar to TRRAP-containing complexes, do not bind DNA directly. We also confirm initial observations from the Henikoff lab that CUT&RUN-seq has a very high signal-to-noise ratio, substantially reducing the cost of sequencing.

3) The experiments implicating both TIP60 and SAGA in the repression of the IRF9 gene are not convincing. This part should be removed from the manuscript as substantial additional work would be required to make this claim convincing. To argue for a direct affect ChIP/PCR experiments on IRF9 are required. Interpretation of the expression changes observed on knock down of SAGA and TIP60 components are complicated by different efficiencies of the knockdowns and by the fact that these components can be components of other complexes and/or function independently in sub-complexes. Finally, different genes require different parts of SAGA for their expression, thus it is likely that different subunits would have different affects on any repression mechanism

Interpreting the effect of siRNA-mediated knockdown of SAGA and TIP60 subunits is indeed complicated by possible confounding factors and we are grateful to the reviewers for considering this point as beyond the scope of the manuscript. As suggested, we entirely removed the paragraph entitled ”*Both TIP60 and SAGA contribute to ISG repression*” from the Results section, along with Figure 7, and the corresponding sentence and information from the Discussion (page 28) and Materials and methods. We kept and modified our discussion of the possible molecular mechanisms by which TRRAP functions as a transcriptional repressor at specific genes (page 28-29, paragraph starting with “*How does TRRAP represses specific ISGs?*”).

4) The authors mention in the methods section that heterologous DNA was used to normalize CUT&RUN experiments but make no reference to this normalization in their figures or explained in the methods. In the presented data it is certainly not explained what "AU" (occupancy levels) corresponds to technically, while IgG controls are seemingly not used as reference point. The numbers presented are extremely variable and it is difficult to judge relative TRRAP binding to the 3 different promoters. If the CUTnRUN works so well, why not performing NGS and get a global view of TRRAP binding on the genome?

We apologize for this confusion. We have now clarified this point and explain in more details how we processed both the CUT&RUN-qPCR and the novel CUT&RUN-seq data in Materials and methods, page 37-38. We also updated the description of the CUT&RUN-qPCR data in the corresponding figures and legends, which are now Figure 6C and Figure 5—figure supplement 1.

For CUT&RUN-qPCR, we used a genomic DNA sample of known concentration in each plate to transform Ct values into DNA quantity values and to calibrate experiments performed on different days and by distinct experimenters. These values represent raw amounts of DNA released after pA-MNase activation, or “TRRAP footprints”, without any further normalization. We thus changed the y-axis label in both figures from “occupancy levels” to “CUT&RUN signal”. In addition, we repeatedly found more TRRAP binding at the *MIR17-HG* promoter than at *IRF7* and *IRF9*, relative to IgG controls. However, we noted that the values obtained for the IgG samples are very low, close to the limit of the detection by qPCR, such that they cannot be used confidently as a reference point for calculating an enrichment ratio. Finally, we addressed the issue of variability by adding more replicates to the control condition, particularly for *IRF7* and *IRF9* (Figure 6C). While merging all experiments, we noticed an error in processing the data for *IRF9* in the original version of the figure, which likely explains why the values were so different between *IRF7* and *IRF9*. This has been corrected and we thank the reviewers for bringing this issue to our attention.

For CUT&RUN-seq, we have now included these data in the revised manuscript and provide a global view of TRRAP chromatin occupancy in HCT116 cells (see our response to comment #2 for details). Here, although we added sonicated genomic DNA from *Drosophila* S2 cells as a heterologous spike-in, as recommended in the original protocol, we did not use it during data analysis. Indeed, for this manuscript, we performed neither a differential nor a quantitative analysis of TRRAP binding (Figure 5). Rather, we focused on the steady-state level of TRRAP binding to promoters and compared it with its effects on the transcriptome measured by our RNA-seq of TRRAP-depleted cells.

5) In addition, a drop in MYC occupancy at MIR17-HG promoter following auxin induction is observed and the authors explain this by a role for TRRAP in stabilizing MYC at its target genes. However, non-specific effects of auxin on CUT&RUN results are not ruled out. Profiling an additional factor that should not be affected by TRRAP depletion would be necessary to validate and increase confidence in the results obtained in Figure 6, where the authors look at the dynamics of ISG regulation by TRRAP over a time course after removing auxin by coupling CUT&RUN to RT-PCR analyses, and to confirm that TRRAP indeed is a direct repressor of IRF9.

We thank the reviewers for raising this point, which we addressed by measuring the effect of auxin on cells lacking AID-tagged TRRAP. For this, we performed a anti-MYC CUT&RUN experiment in the parental, TIR1-expressing HCT116 cell line used for CRISPR-Cas editing of *TRRAP*. We reasoned that finding a factor that is not affected by TRRAP, which can interact with many different transcription factors, might be challenging. We observed that treating HCT116 cells with auxin for 12 hours had no effect on MYC binding to the *MIR17-HG* promoter, contrasting with the decrease observed in auxin-treated AID-TRRAP cells (Figure 5—figure supplement 1A). We conclude that the effect of auxin on MYC binding is specific to TRRAP depletion, validating our CUT&RUN results. These data are presented in a much-revised version of what is now Figure 5—figure supplement 1A and described in the paragraph from the Results section that describe our implementation of the CUT&RUN procedure (page 17-18). We are grateful to the reviewers for suggesting this important control, which clearly strengthens the manuscript.